# Diversity of major histocompatibility complex of II B gene and mate choice in a monogamous and long-lived seabird, the Little Auk (*Alle alle*)

**Katarzyna Wojczulanis-Jakubas**[1]*, **Brian Hoover**[2], **Dariusz Jakubas**[1], **Jérôme Fort**[3], **David Grémillet**[4], **Maria Gavrilo**[5¤], **Sylwia Zielińska**[6], **Magdalena Zagalska-Neubauer**[7]

1 Department of Vertebrate Ecology and Zoology, Faculty of Biology, University of Gdansk, Gdansk, Poland, 2 Farallon Institute, Petaluma, California, United States of America, 3 Littoral, Environnement et Sociétés (LIENSs), UMR 7266 CNRS – La Rochelle University, 17000 La Rochelle, France, 4 Excellence Chair Nouvelle Aquitaine - CEBC UMR 7372 CNRS, La Rochelle Université, Villiers-en-Bois, France & FitzPatrick Institute of African Ornithology, University of Cape Town, Rondebosch, South Africa, 5 National Park Russian Arctic, Archangelsk, Russia, 6 Department of Molecular Biology, Faculty of Biology, University of Gdansk, Gdansk, Poland, 7 Department of Behavioural Ecology, Faculty of Biological Sciences, University of Wroclaw, Wroclaw, Poland

¤ Current address: Arctic and Antarctic Research Institute, Saint-Petersburg, Russia
* katarzyna.wojczulanis-jakubas@ug.edu.pl

**Data Availability Statement:** All sequences obtained, which are the core of the study, are

## Abstract

The major histocompatibility complex (MHC) plays a key role in the adaptive immune system of vertebrates, and is known to influence mate choice in many species. In birds, the MHC has been extensively examined but mainly in galliforms and passerines while other taxa that represent specific ecological and evolutionary life-histories, like seabirds, are underexamined. Here, we characterized diversity of MHC Class II B exon 2 in a colonial pelagic seabird, the Little Auk (or Dovekie *Alle alle*). We further examined whether MHC variation could be maintained through balancing selection and disassortative mating. We found high polymorphism at the genotyped MHC fragment, characterizing 99 distinct alleles across 140 individuals from three populations. The alleles frequencies exhibited a similar skewed distribution in both sexes, with the four most commonly occurring alleles representing approximately 35% of allelic variation. The results of a Bayesian site-by-site selection analysis suggest evidence of balancing selection and no direct evidence for MHC-dependent disassortative mating preferences in the Little Auk. The latter result might be attributed to the high overall polymorphism of the examined fragment, which itself may be maintained by the large population size of the species.

## Introduction

Producing two major classes of cell surface receptors (class I and class II), the polymorphic genes of the major histocompatibility complex (MHC) play an essential role in the adaptive

available in GenBank (Accession numbers: GenBank OR909382-OR909480).

**Funding:** Our study was supported by grants from Poland through the Polish Ministry of Science and Education (Juventus Plus 0470/P01/2010/70), samples on Greenland were collected based on the ADACLIM project (No 388), funded by the French Polar Institute. (IPEV) and LIAK&CC project (project Marie Curie IEF number 273061) funded by the European Commission. Sampling conducted in the specially protected area of the Franz-Josef Land State Federal Refuge, within the framework of the joint international expedition of National Park Russian Arctic and a National Geographic 'Pristine Seas Expedition FJL 2013', was supported by National Geographic, Blancpain and Davidoff Cool Water. The funders had no role in study design, data collection and analysis, decision to publish, or preparation of the manuscript.

**Competing interests:** The authors have declared that no competing interests exist.

**Abbreviations:** cDNA, complementary deoxyribonucleic acid; DNA, deoxyribonucleic acid; dNTPs, deoxynucleotide triphosphates; MHC, major histocompatibility complex; PBS, peptide-binding sites; PCR, polymerase chain reaction; RNA, ribonucleic acid.

immune function of vertebrates [1–3]. Since MHC receptors are encoded by individual alleles that respond only to a fixed number of antigenic peptides, more diverse MHC genotype provide better protection from a broad range of pathogens and is advantageous for disease resistance [4, 5]. Therefore, high levels of MHC polymorphism in a population often reflect selection pressure from pathogens [6–10].

Several genetic mechanisms have been hypothesized to maintain MHC diversity, with three most frequently evoked: heterozygote advantage [11], rare-allele advantage [12, 13] and fluctuating selection [14–16]. These mechanisms represent forms of balancing selection that favour the existence of multiple alleles within a population [17]. Since MHC genotypes may be phenotypically detectable [18–22], MHC polymorphism, especially MHC class II [23], may be behaviourally maintained through strategies, such as mate choice, wherein the genotypes of both parents influence the immunocompetence of the offspring [24]. In this context, the genetic compatibility hypothesis assumes that individuals may produce MHC-heterozygous offspring if they mate disassortatively in respect to their own MHC profile (i.e. partner's genes are supposed to be sufficiently different or complementary to own genotype) [20, 25].

While the avian MHC complex has been extensively examined, the majority of these studies has been performed on specific ecological/taxonomical groups such as galliforms or passerines. In the context of mate choice and heterogenous distribution of pathogens in the environment, however, other avian taxa that represent distinct ecological and evolutionary life-histories, may experience different selection pressure on MHC polymorphism [26, 27]. Therefore, any general interpretation of the mechanisms and phylogenetic origins of MHC diversity in wild bird populations should ideally be based on a wide array of avian taxa [28]. Consequently, there is a strong need to characterize MHC patterns across different avian species.

Due to habitat preferences and some specific life-history traits, seabirds represent an interesting group to examine the functioning of the immune system, including MHC (e.g. [23, 29]). Seabirds spend most of the year in the marine environment where they are exposed to an array of specific pathogens and endo- and ectoparasites [30–32], and that may differentiate them from other avian groups. In addition, many seabirds breed colonially in dense aggregations that can promote elevated transmission rates of pathogens, and the strength of selection on MHC diversity (especially in class II genes) has been shown to increase with increased coloniality and migratory behavior in birds [33]. Recent outbreaks of avian influenza have had particularly strong effects on (colonial) seabirds [34, 35], highlighting the need for a deeper understanding of the functioning immune system in this group. Finally, seabirds form long-term pair bonds that are critical for the successful incubation and provisioning of the brood, and thus represent a promising avenue for MHC-dependent mate-choice studies [36–39]. In this study, we focus on the Little Auk, also knowns as the Dovekie (*Alle alle*), to characterize their MHC diversity and to test whether MHC variation could be maintained in this species through MHC-dependent mating preferences.

The Little Auk is a small (130–180 g) pelagic seabird that breeds exclusively in the High Arctic [40], with wintering areas also located mainly in sub-Arctic and Arctic zones [41]. This makes the species particularly intriguing in the context of MHC studies as polar marine ecosystems are characterized by relatively low levels of pathogen abundance [42–46]. Such environmental conditions could be expected to relax pathogenic selection on the MHC diversity in Little Auks. However, Little Auks also breed colonially in high densities that may facilitate pathogen transmission [27] (for example, via direct contact during sexual or aggressive interactions [47]), which in turn could increase selection pressure on MHC alleles, therefore increasing MHC diversity. These contrasting scenarios predict opposing patterns of MHC diversity in the Little Auk and raise the need to understand the role of MHC in mating preferences and population biology of this species. In this study, we therefore analyzed MHC alleles

across three main breeding locations (comprising two morphologically distinguishable subspecies) to recognize MHC diversity of the species. Since selecting a mate is a critical decision with long-term fitness consequences in Little Auks (long-term pair bonds, social and genetic monogamy [48], preference of partners with particular phenotypic traits [49]), and MHC II genotypes are known to mediate mate-choice decisions [24], we also tested the hypothesis that Little Auks exhibit MHC-dependent mating preferences.

## Materials and methods

### Samples collection

To characterize MHC diversity in Little Auks, we collected small blood samples (ca. 20 μL of whole blood collected from underwing vein and preserved in 90% ethanol) from 140 birds originating from three breeding colonies: Hornsund, Spitsbergen (77˚ 00'N, 15˚ 33'E, n = 99 in 2014); Ukaleqarteq (Kap Höegh; in 2013), East Greenland (70˚ 43'N, 22˚ 38'W, n = 12 in 2013); and Tikhaya Bay, Hooker Island, Franz Josef Land (80˚20'N; 52˚49'E, n = 29) (Fig 1). Spitsbergen and Greenland colonies are inhabited by the nominative subspecies *Alle a. alle*, while Franz Josef Land is inhabited by the subspecies *Alle a. polaris*. The two subspecies are distinguished based on body size differences (the nominative subspecies is smaller) but are genetically very similar when assessed using neutral genetic markers [50]. The three colony locations sampled represent the most important breeding aggregations of the species (after the Thule area in Northwest Greenland, [51]). The birds sampled in Spitsbergen were breeding birds (44 pairs plus single male–sexed molecularly), captured in the nest while incubating. The birds from Greenland and Franz Josef Land were also breeding adults but represented random individuals captured in mist-net or foot-loop traps during the chick rearing period (breeding status was established based on the gular pouch containing food for the chick). The sex of the birds sampled from Greenland and Franz Josef Land was not established but given the randomized capture protocols and similar colony attendance pattern for males and females, the sex ratio of these samples should be balanced. In summary, samples collected from all three colonies were used to characterize the general pattern of MHC diversity in the species, and the samples collected form the Spitsbergen colony were used to identify sex specific patterns of

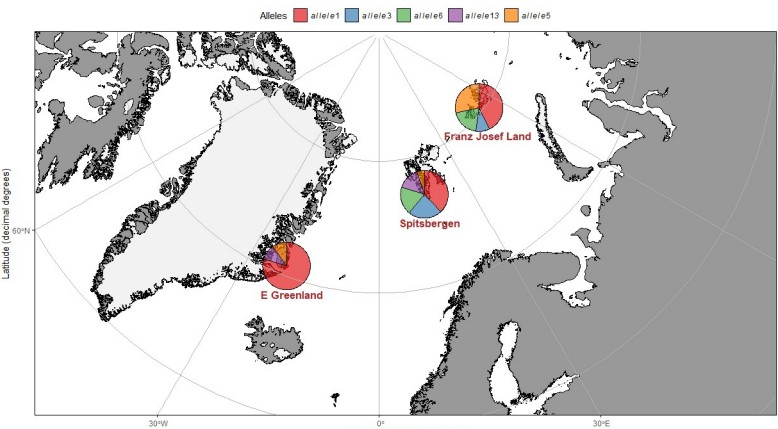

**Fig 1. The relative frequency (bars) of five the most frequent alleles (overall observed in >10 individuals) across the three sampling locations representing three little auk breeding colonies.** The map has been prepared in *ggOceanMaps* [101] and *scatterpie* [102] in packages R software 3.6.3 (R Core Team 2020). Map source: Natural Earth (public domain): http://www.naturalearthdata.com/.

MHC diversity and potential MHC-dependent mating preferences within known breeding pairs.

All birds were individually marked with ornithological metal rings to avoid double sampling of the same individual. Eight birds from Greenland population were sampled *post-mortem* as subjects of another project (ADACLIM no: 388 of JF and DG). From those individuals, the spleen was collected immediately (<5 min) after euthanasia. The spleen was then preserved in RNAlater solution (Qiagen) and stored at -20˚C until the laboratory analysis. We used the spleen samples to amplify and sequence complementary DNA (cDNA), to control and validate the expression status of MHC class II B sequences obtained from DNA samples (see the details below).

## Ethics statement

All the samples were collected following rules for the use of animals, as specified in the guidelines of the Association for the Study of Animal Behaviour. All blood sampling and bird euthanasia was performed under permission issued by the Norwegian Animal Research Authority, the Governor of Svalbard (the Spitsbergen population) and Government of Greenland (Ministry of Fisheries, Hunting and Agriculture, the Greenland population): 2007/00150–9, 2007/00150–11, 17/00663–2, 17/00663–7; 2011–047447, and permits from Russian Ministry of Higher Education (the Franz Josef Land population): 71 from 03.06.2013.

## DNA and RNA extraction

DNA was extracted from the blood using the kit Blood Mini (A&A Biotechnology, Poland). RNA was extracted from the spleen tissue with the use of MasterPure™ Complete DNA and RNA Purification Kit (Epicentre), according to manufacturer's protocol. For tissue homogenization PRO DPS-20 Multi-Sample Dual Processing Homogenizing System (mechanical homogenization– 16000rpm for 60 sec—2 cycles, ultrasonic homogenization– 75% input amplitude for 30 sec) was applied. Extracted RNA was portioned and stored at -80˚C for further use.

## Molecular sexing

For sexing (Spitsbergen population only), we performed PCR using the avian sexing primer pair P2 and P8, following the PCR procedure described in [48]. To evaluate PCR products, we applied standard electrophoresis procedure by dyeing the fragments with Midori Green (Nippon, Europe) and examining bands under UV light using a 2.5% agarose gel.

## Primer development

To characterize MHC diversity in the Little Auk, we identified and sequenced the second exon of MHC class II B genes, which is the most polymorphic segment of the peptide-biding region (PBR) [52]. First, we designed and tested a set of primers (Table 1 and Supplementary materials: S1 Table), based on available sequences for closely related alcid species: the Razorbill (*Alca torda*) (Accession Number: EU326270.1, EU326269.1, EU326268.1), the Common Guillemot (*Uria aalge*) (Accession Number: EU326278.1, EU326276.1, EU326275.1EU326274.1) and the Atlantic Puffin (*Fratercula arctica*) (Accession Number: EU326267.1, EU326267.1, HQ822507.1, HQ822494.1). In the next step, the designed primers along with universal ones (C20, B21, [53]) were used in vectorette PCR, to obtain longer exon sequences covering the region, where we intended to locate the final primers for genotyping (Table 1 and Supplementary materials: S1 Table). In the vectorette PCR, we used total genomic DNA from the eight

**Table 1. Sequences of primers that identify exon 2 MHC in the Little Auk, used for final MHC genotyping.**

| Primer name | Sequence 5'—3' |
| --- | --- |
| AukLz1 | ATGTCTGCMCGAGCAGGGWA |
| AukRz3 | CCRGGGCTTGGCTTGTGCCTG |
| AukL0709 | GCCCTTGGGCACTGGTGCA |
| AukR0709 | TCCGTTCTGCATCACCTCCGT |
| AukL0809 | GCAGGAGGATGCTGTGCAGGGA |
| AukR0809 | TCACCAGCACCTGGTAGGTCCAGTC |

individuals digested with four restriction enzymes to generate vectorette-ligated DNA templates.

For vectorette PCR, we adopted the modified vectorette PCR protocol from [53] as it allowed us to obtain information on poorly known DNA fragments of interest, and has been applied successfully in MHC research (e.g. [54–56]). In this technique, one primer (so called "internal") is designed based on the available sequence, and the second primer is universal (C20 and B21 used as forward or reverse). This approach allows the 'unknown' part of the sequence to be extended in one direction and amplified (contrary to standard PCR, which requires two specific primers for amplification). Therefore, the technique provides an opportunity to design more precise, final primers for focal fragments.

First, to construct vectorette libraries, we used ca. 5 μg portion of genomic DNA which was digested with 10 U of enzymes: MunI, XapI, EcoRI, Bsu15i REs (Fermentas) at 37°C for 5 min (other steps, including ligation, were performed according [53]). Then, we have performed vectorette PCRs, where 15 μl of reaction mixture contained 7.5 μl QIAGEN Multiplex PCR Master Mix, 0.15 μM of the specific (AukLz1 or AukLz2/AukRz1, Table 1 and Supplementary materials: S1 Table) and C20 vectorette [53] primers, and 0.5 μl of vectorette-ligated digested DNA. The PCR touchdown was as follows: 95°C/15 min, 5×(94°C/30 s, 64°C/30 s, 72°C/60 s), 5×(94°C/30 s, 60°C/30 s, 72° C/60 s), 20× (94°C/30 s, 58°C/30 s, 72°C/60 s), 72°C/3min. In the second nested vectorette we used PCR 0.5 μl of product of the first PCR as template. 30 μl of PCR mixture contained 2 μL of 10× PCR buffer with $(NH_4)_2SO_4$, 2 mM $MgCl_2$, 1 μM of each primer, 0.2 mM of each dNTP and 1 U of Taq polymerase (Fermentas). In this PCR we used one specific primer (Alal1L/Alal2R/Alal3R, Supplementary materials: S1 Table) and B21 vectorette primer [53], PCR scheme was as follows: 94°C/3 min, 30× (94°C/30 s, 58°C/30 s, 72°C/60 s), 72°C/3 min. The second PCR products were run on a 1.5% agarose gel and visible fragments (ca. 300 bp) were cut off from the gel and purified with MiniElute Gel Purification Kit (Qiagen). These fragments were Sanger-sequenced with specific vectorette primers. The obtained sequences were then compared to the NCBI nucleotide resources with nblast (Nucleotide BLAST, NCBI). Based on the sequencing results, we selected fragments that aligned to MHC II B exon 2 and corresponded to sequences known from closely related auk species. Ultimately, we were able to select final primers: AukLz1 and AukRz3 for MHC II B second exon genotyping (Table 1). The PCR product of additional pairs of primers (AukR0709, AukL0709 and AukL0809, AukR0809; Supplementary materials: S1 Table) was also identified as a region of interest, i.e. the second and partial third exon of the MHC class II gene. However, generated fragments were too long to use freely with Ion Torrent technology.

## Primers validation

The complementary DNA (cDNA) samples were used to generate sequences which were regarded as a control of the reliability of functionality of MHC fragments amplified with

designed specific primers. Fragments obtained from cDNA samples represent putative, functional alleles, not non-functional pseudogene. Therefore, in the following step we confronted MHC sequences for DNA and cDNA samples generated with use of final primers: AukLz1 and AukRz3 (Table 1). Finally, before performing the next-generation sequencing of all samples, to confirm target sequence (MHC II, exon 2 fragment), we also checked the PCR product (of DNA and cDNA samples) by Sanger sequencing in two directions with both final PCR primers. For validation, the sequenced fragments were aligned with use of the Basic Local Alignment Search Tool (BLAST).

To obtain cDNA we used RNA template (n = 8) and performed reverse transcription to generate cDNA. In the first step (pre-processing) we added 2μl of RNA template (in total around 50ng) with 0,5 μl of 10 mM primer AukRz1 (degenerate primer, Table 1) and 10,5 μl of nuclease-free treated water and incubated in 65˚C for 5 min, according to manufacturer's protocol. Then, we cooled the sample to room temperature. In the next step, we added 4 μl of 5x Reaction Buffer (part of RevertAid™ M-MuLV) to the preprocessed sample with 2 μl of mixed dNTP (10 mM, Thermo Scientific) and 1 μl of reverse transcriptase RevertAid™ M-MuLV, Thermo Scientific. We incubated the mixture for 1h at 42˚C according to manufacturer's protocol and the reverse transcriptase was inactivated for 10 min at 70˚C.

Subset of DNA samples and all cDNA samples were used as a matrices in an amplification reaction in 21μl volume with the use of QIAGEN Multiplex PCR Kit (10 μl of 2x QIAGEN Multiplex PCR Kit, 2 μl of each 10 mM final primer and 6 μl of water). For each reaction we used a 1μl of template. We performed all reactions using a thermal cycler (Eppendorf) under the following conditions settings: 94˚C for 15 min for initial denaturation followed by 34 cycles of 94˚C for 30 sec, 71˚C for 90 sec and 72˚C for 90 sec, with additional 72˚C for 10 min at the end. Obtained products were visualized by gel electrophoresis, Sanger sequenced and aligned.

## Next generation sequencing

We genotyped 298-bp long amplicons of the Little Auk MHC class II B second exon with Ion Torrent Personal Genome Machine (PGM, Life Technologies, Carlsbad, CA, USA) in the Laboratory of Molecular Biology Technics at Adam Mickiewicz University in Poznań (Poland). First, we performed PCR with so-called fusion primers comprised of Ion Torrent adaptors (A adaptor and P1 adaptor), a 10-bp barcode, and our designed pair of specific final primer (for PCR conditions see the section above). The unique barcodes were used to assign amplicons to individuals (Supplementary materials: S2 Table). In total, a combination of 16 tagged forward primers and 16 tagged reverse primers was used to amplify individual samples (with different combinations of tags and primers). Two independent PCRs were performed on two sets of samples: 140 (all) and 126 individuals (randomly selected of the 140). Following amplification we prepared two sets of individuals pooled equimolarly and sequenced independently to estimate genotyping error. To check the quality of both sequencing data sets we used *Rqc Bioconductor* package [57].

## Data filtering

The processing of raw data was conducted with AmpliSAT, which provides a wide range of online tools for amplicon analysis, and is also appropriate for technology that generates single-ended reads such as Ion Torrent ([58]; http://evobiolab.biol.amu.edu.pl/amplisat/index.php). Following the pipeline AmpliSAT manual, we applied the AmpliCLEAN, AmpliCHECK, AmpliSAS, AmpliCOMPARE, and AmpliCOMBINE tools. These tools provided a standardised approach for identifying and filtering reads, variants and amplicons according to their similarity, coverage and frequency. We removed reads with a lower average Phred quality

score ($< 30$) and reads with anomalous length (either too short or too long, as identified by AmpliCLEAN). To filter out the putative alleles from artifacts we used default Ion Torrent parameters: 0.5% substitution error rate, 1% indel error rate and minimum per amplicon frequency of 1%. We also removed sequencing errors [59] and chimeras (the most abundant ones) [58, 60] setting 5% frequency threshold of substitution (AmpliCHECK). To compare replicated genotyping results first, we combined multiple genotyping results (AmpliCOM-BINE) and then checked for variants present in one or both compared files (AmpliCOM-PARE). For identified alleles, variant depth and other statistics were summed together from both runs of sequencing, and sequencing errors were removed based on their low frequency and/or absence in one set of individuals [54, 60].

In total, we successfully sequenced 140 individuals (including those for which we obtained cDNA samples). Of these, 126 individuals were sequenced twice. Individuals that were sequenced only once were included in analyses if they provided at least 200 reads per allele. All sequences exhibited the correct size (298 bp) with no stop codons and no frame-shift mutations which allow exclude putative pseudogenes. They also correspond to sequences obtained for cDNA samples. Therefore, obtained MHC class II exon 2 fragments were considered putative alleles and submitted to GenBank (Accession numbers: OR909382-OR909480).

## Genetic analysis

We used the DNAsp software [61] to calculate basic polymorphism statistics, including the overall number of segregating (variable) sites, nucleotide diversity ($\pi$), Watterson's estimate of the population mutation rate ($\theta\omega$), and the average number of nucleotide differences among alleles ($k$). We used Tajima's D test [62] within DNAsp to analyse the frequency distribution of variable sites, based on a window size of 25 nucleotides and a step size of 5 nucleotides. Departure from neutral selection at MHC Class IIB in Little Auks was tested within putative PBS and non-PBS regions, which were defined using [63] and previous seabird genetic studies, i.e. [29]. We tested for evidence of positive selection across the alignment of exon 2 sequences using the HyPhy package [64] in the webserver www.datamonkey.org, and applied the Random Effects Likelihood method to investigate site-by-site evidence of positive selection. As recombination can overestimate the degree of positive selection [65], we ran the Genetic Algorithm for Recombination Detection module within the HyPhy package to test for significant evidence of recombination breakpoints. Finally, we used the software MEGA 4.0 [66] to test synonymous vs. non-synonymous substitution rates within codons presumably representing the protein-binding region within each amplicon (PBR) [67], using the Modified Nei-Gojorobi method with Jukes-Cantor correction.

## Mate-choice analysis

We used four distinct genetic metrics to test whether Little Auks select mates on the basis of MHC compatibility: i) differences in heterozygosity; ii) MHC-band sharing coefficients [68]; iii) additive amino acid differences; iv) maximum amino acid difference. Heterozygosity differences were calculated as the overall difference in MHC heterozygosity between each breeding pair, and can only be used to test whether disassortative mating occurs within a population (e.g. whether males or females are on average significantly more heterozygous at MHC II B than their partners). Band-sharing represents the proportion of unique MHC alleles shared within a breeding pair, and is calculated as twice the number of alleles shared within a pair, divided by the sum of unique alleles occurring within both individuals [($2F_{ab}/(F_a+F_b)$, as per [67]]. Amino acid differences represent the number of amino acid substitutions occurring between pairs of different MHC alleles, and were quantified here as either the sum of all the

differences between a pair at both loci, or the maximum difference observed between loci. We assessed these amino acid substitutions within two different contexts; i) using only the substitutions occurring within PBR regions of the MHC II B sequence, and ii)) using substitutions occurring throughout the entire fragment of the exon sequence. For each of the four criteria listed above, we used Monte Carlo randomization tests [69] to determine whether average observed mating patterns for each metric were significantly different from simulated expectations of random mating. To obtain random mating distributions, we resampled males and females from the study population without replacement and formed new random pairs, performing 10,000 iterations of this randomization procedure. We then compared the mean observed values for each metric (i.e. the values we calculated for naturally occurring pairs) with the distribution of randomly calculated means. We determined statistical significance by comparing the observed mean of the breeding pairs with the 2.5% and 97.5% tail of the randomized distribution of means, and Bonferroni-adjusting the alpha value by the number of randomization tests (0.05/6 = 0.008). For each metric, we assessed the likelihood of error within Monte Carlo randomization tests by iteratively decreasing the sample size and examining the effect of sample size on the consistency of obtaining a given result (for further explanation, see [70]). We performed all statistical analyses in R 3.6.3 software [71].

## Results

### MHC diversity and selection

We characterized a high level of genetic diversity at MHC II B exon 2. We sequenced 298 bp fragments that contained 99 distinct alleles across 140 individuals. Within this set of individuals, eight samples were also used for cDNA sequencing (spleen samples). We did not perform any formal analysis but examining alleles from the eight individuals, we could confirm the expression status of identified MHC fragments (putative alleles), generated by primers, thus MHC diversity we quantified in the Little Auk is reliable.

A maximum of four alleles was documented within an individual, indicating at least two loci, i.e. the presence of a duplicated MHC II gene fragment, which is consistent with findings in other alcid species [72, 73]. On average two alleles per individual were recorded, both within the total sample pool and separately for each population. Overall, four individuals had 4 alleles (3%), 39 individuals had 3 alleles (28%), 56 individuals had two alleles (40%), and 41 individuals had one allele (29%).

Out of 99 alleles identified from nucleotide sequences 54 were present only in a single individual, 19 other alleles were present in two individuals, and remaining 26 alleles were observed in three or more individuals (see Supplementary materials: S3 Table). The number of alleles identified per population was: in Spitsbergen—75 alleles (64% of these were unique to the population, of n = 99), in Greenland—20 alleles (35% unique, of n = 12) and in Franz Josef Land—36 alleles (44% unique, of n = 29). The frequency of the five most frequent alleles, i.e. observed in >10 individuals, in each of the sampled colonies is presented on Fig 1. Obtained alleles translated into 89 amino acid sequences. Genotyped alleles exhibited a similar skewed distribution in both sexes, with the four most commonly occurring alleles representing approximately 35% of allelic variation (Fig 2). Basic polymorphism statistics are reported in Table 2. We found a significant excess of non-synonymous over synonymous substitutions for those codons presumably comprising the peptide binding site (PBS)-coding region ($\omega$ = 2.404; Z-test, p = 0.004), but not for the non-PBS regions ($\omega$ = 1.257; Z-test, p = 0.132), suggesting evidence of a positive selection. A Bayesian analysis of site-by-site selection across an alignment of exon 2 sequences indicated three sites undergoing a positive selection and one site

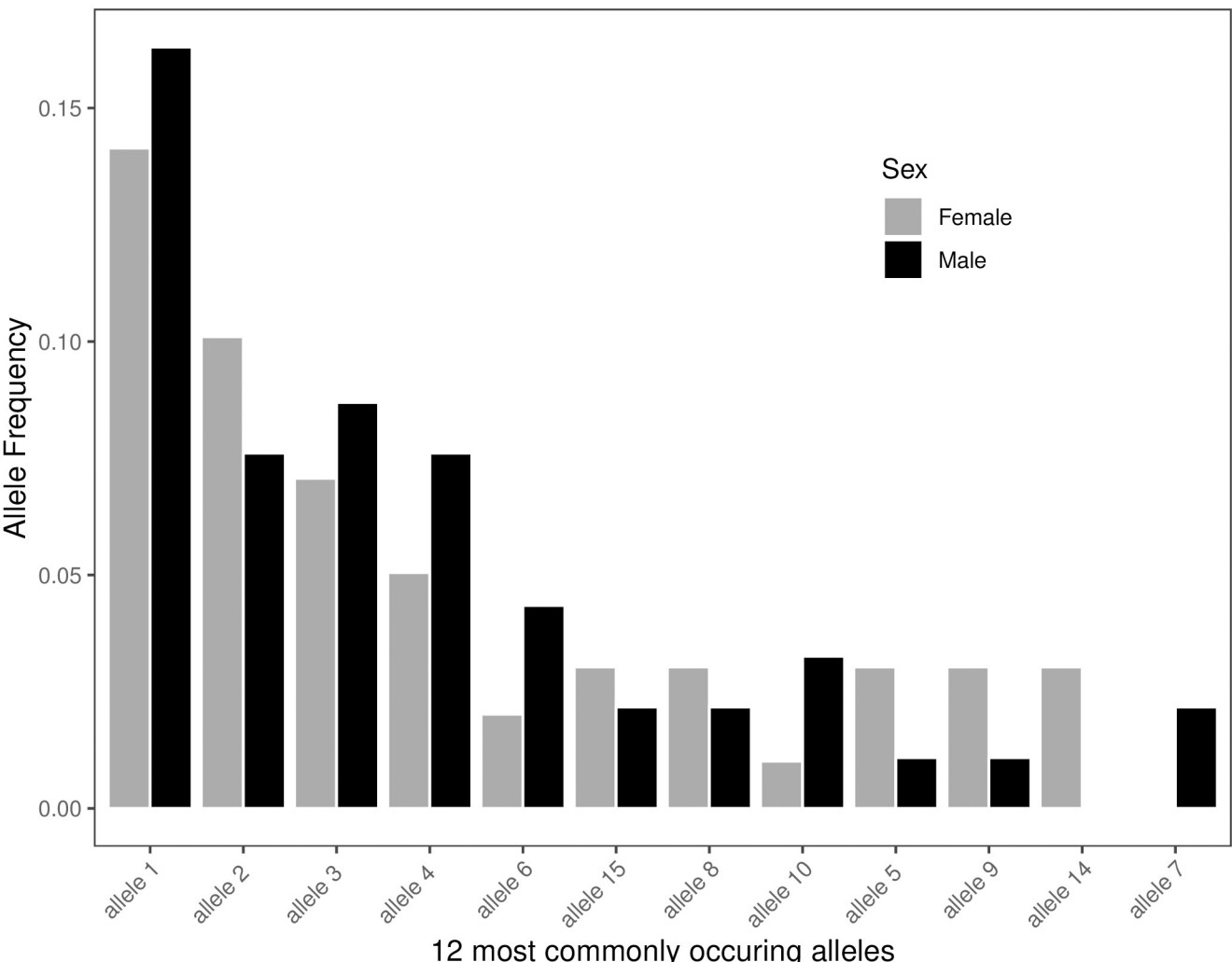

**Fig 2. Distribution of allele frequencies at genotyped MHC IIB loci in the little auk in respect to sex (Spitsbergen colony only).** Only alleles found at frequencies > 2%, in either sex, are depicted.

**Table 2. Polymorphism statistics at genotyped MHC II B loci in Little auks.** Diversity statistics: the overall number of nucleotide sequences found (h), the overall number of segregating sites (S), the total number of mutations (*Eta*), the nucleotide diversity estimator ($\pi$), the Watterson's estimator of the population mutation rate ($\theta_W$) and the average number of nucleotide differences among alleles K).

| Diversity statistic | Value |
|---|---|
| h | 99 (69) |
| S (*Eta*) | 87 (89) |
| ($\pi$) | 0.063 |
| ($\theta_W$) | 0.069 |
| K | 16.67 |

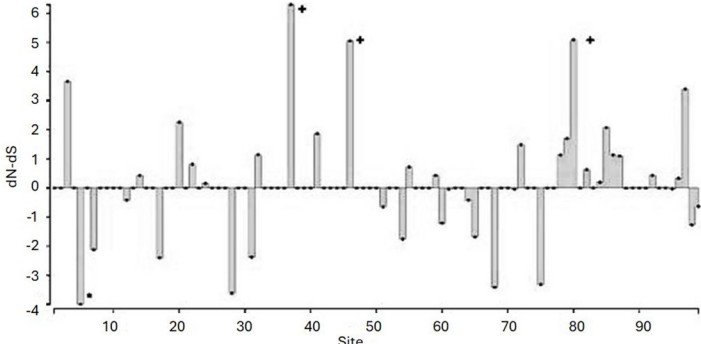

**Fig 3. Bayesian analysis of site-by-site positive selection across an alignment of exon 2 sequences from two MHC class II B gene duplicates in the little auk.** Codons displaying decisive evidence of positive selection (Bayes factors >100) are indicated by crosses. Sites showing decisive evidence of negative selection are indicated by asterisks.

undergoing a negative selection (Fig 3, Bayes Factor 100), while Tajimas D analysis revealed sites with a significant excess of high- frequency segregating sites (S4 Table).

## Disassortative mating

We found no evidence of non-random mating based on overall differences in MHC II heterozygosity, average amino acid dissimilarity, or maximum amino acid dissimilarity using the whole fragment region (Fig 4) or PBS regions (S1 Fig). We found a significant effect of MHC-based assortative mating on shared alleles, suggesting that breeding pairs shared alleles in common more often than would be randomly expected (p = 0.016). However, this effect may be biased by the inability to distinguish between duplicated loci. Overall, insufficient evidence is available to conclude the presence of MHC-based mating effects.

## Discussion

Our study revealed a high level of MHC class II B diversity within Little Auks, one of the highest described in non-passerine birds. In 140 screened individuals, we detected 99 different alleles in MHC II B exon 2, with an average 16.2 nucleotide differences between alleles. Four alleles were dominant in the allele distribution (Fig 1) but most genotypes represented rare alleles. We are confident the high number of detected alleles is accurate as we performed PCR in several independent sets, samples were sequenced independently twice, obtained sequences did not resemble chimeras, and were regarded as reliable alleles if present in a high number of reads (minimum 200, in two runs). Additionally, we have applied restricted criteria to sort out sequences. We also used cDNA to exclude putative alleles from the analyses (i.e., a with no stop codons, deletion or frameshift mutations).

The high number of MHC alleles could be generated by a number of different processes such as point mutation, recombination and gene conversion [74–76]. Generated alleles might be then preserved and maintained by adaptive forces such as balancing selection [74–76]. A high genetic diversity is an inherent feature of MHC genes (including class II) that has been reported in numerous avian species. Especially for passerines, the reported MHC loci and alleles can reach very high numbers, e.g. Sedge Warbler (*Acrocephalus schoenobaenus*) [77], Great Tit (*Parus major*) [78], Collared Flycatcher (*Ficedula albicollis*) [54]. In non-passerine birds the highest allelic MHC diversity has been reported in the Eurasian Coot (*Fulica atra*; 3 loci, 265 alleles) [79], the Greater Flamingo (*Phoenicopterus roseus*; 2 loci, 109 alleles) [80], and

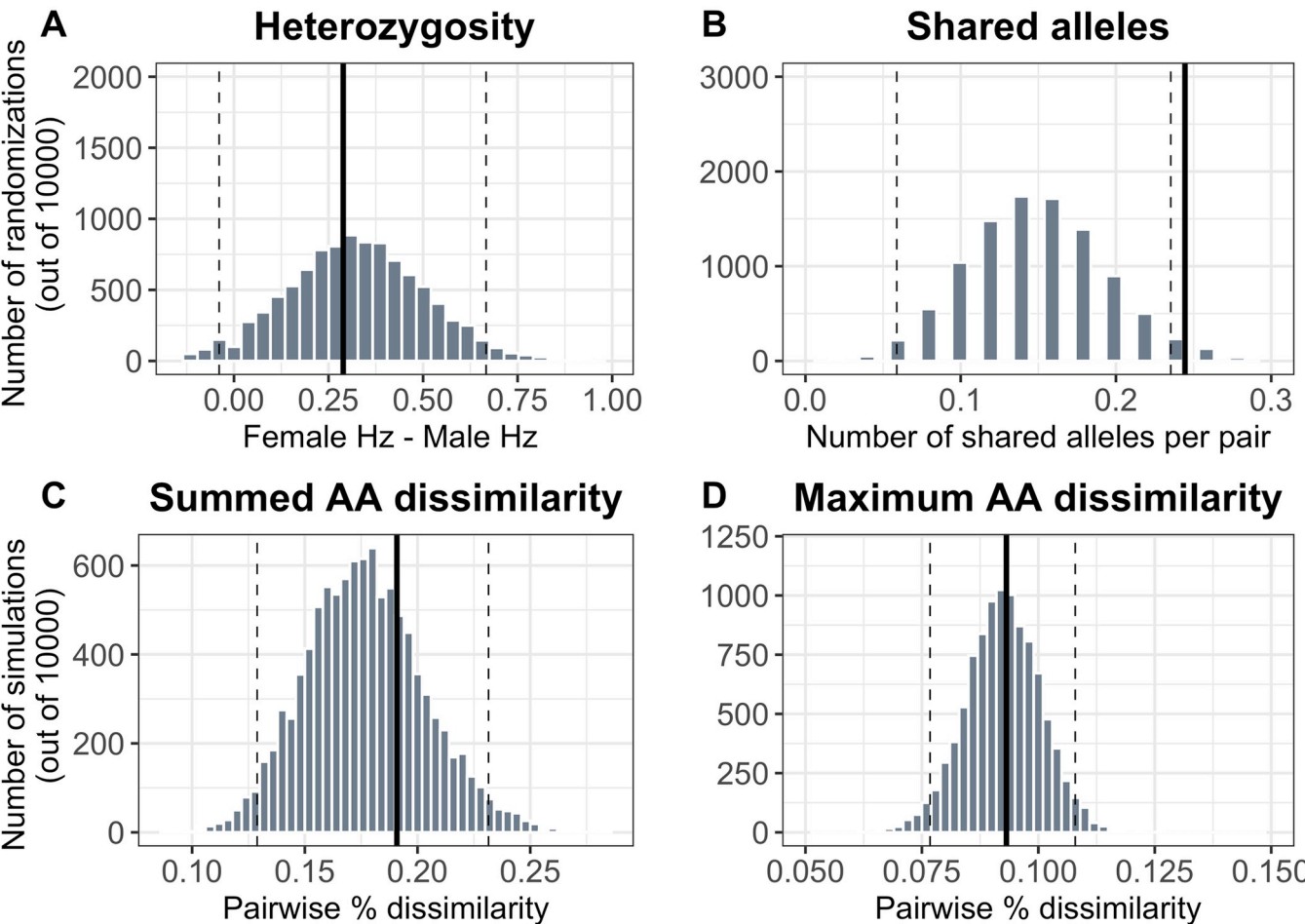

**Fig 4.** A-D. Results of randomization tests testing for MHC-based non-random mating. No significant effects of MHC genotype were found when assessing differences in heterozygosity (4A) or dissimilarities in amino acid composition (4C–4D). Little Auks exhibited a significant assortative mating effect based on shared alleles (4B), which may be attributed to the frequency of the most commonly genotyped allele in the population and the inability to discriminate between locus-specific alleles. Dashed vertical lines denote 95% of random distribution, whereas the solid vertical lines indicate the observed mean value. An observed value located within the 95% random distribution indicates a random mating pattern, whereas an observed mean located outside the 95% range on the right side indicates assortative mating.

the Lesser Kestrel (*Falco naumanni*; 1 locus, 103 alleles) [81]. With 2 loci per individual and a total of 99 identified MHC alleles, the Little Auk polymorphism is lower than these examples, however it remains one of the highest values reported among non-passerines and represents the highest polymorphism value detected so far among seabirds.

This high MHC allele diversity in the Little Auk is quite surprising given the low pathogen-abundance in the Arctic environment [42–46]. For example, no blood parasites have been detected in the Little Auk [82]. However, many MHC loci class I (but not class II) have also been detected in another polar (Antarctic) seabird, the Blue Petrel (*Halobaena caerulea*) [29]. It has been then suggested that the reason for this is that blue petrels are a colonial species that reuse underground nests, which might favor the horizontal transmission of pathogens, even if the pathogens are not that frequent in the polar environment [29]. The Little Auk is also a colonial species, yet Little Auks nest above ground on cliffs or slops, and the horizontal transmission of pathogens in this Arctic environment has not been investigated. Thus, the high degree of MHC polymorphism that we identified in Little Auks may represent an ancestral

polymorphism that is enhanced by a large effective population size, both in the past [51] and at present [83], and subsequently maintained by high gene flow among the breeding colonies [50, 84].

MHC diversity in the Little Auk might be greater than we identified in our study, as we sequenced samples from three breeding locations only. Although the sampled colonies represent a large portion of the breeding area for this species, they certainly do not cover the whole range and some other colonies may be genetically distinct (due to their isolation and/or habitat). High frequency of unique alleles in each population and inter-population differences in the frequencies of the most common shared alleles (Fig 1) suggest there is some divergence between breeding populations and/or adaptation to local conditions (pathogen pressure, population size, etc). Although we are not able to fully examine the inter-population differences in MHC diversity (due to limited population coverage and the small number of sampled individuals), our results suggest biogeographical patterns of MHC diversity within the Little Auk population that are worthy of further examination.

In some seabird populations, MHC polymorphism has also been suggested to result from ancient gene duplication events that led to a functional genetic divergence between duplicated loci [37]. Gene duplication at MHC II B appears to be a conserved trait within many seabird clades, including Procellariformes [38, 85] and alcids [72, 73]. However, in most studies it remains unclear whether such duplicates are functionally identical or could be duplicated loci that have evolved and are expressed independently. While locus-specific MHC II B genotyping was outside the focus of this study, gene duplicates that have diverged and function independently can conceivably maintain high MHC polymorphism [86].

We also found some evidence of positive and balancing selection within Little Auk MHC. As reported in a previous study [33], the strength of balancing selection on MHC class II genes in birds is higher in colonially breeding and migratory species. It may be linked to elevated transmission rates of pathogens in species breeding in dense aggregations, or/and exposure to a more diverse array of pathogens and parasites in migratory species [33]. While we did not investigate gene expression in this study, our evidence supports the idea that the identified alleles are functional and expressed (e.g., [27, 87, 88]). It is further unlikely that the genotyped region represents pseudogenes as those are rarely reported for non-passerines [89–91]. Thus, although we did not examine fitness consequences (e.g. life-time reproductive success) associated with MHC II B polymorphism, in the absence of strong mate choice effects, the high MHC II B diversity we found may be maintained by a combination of pathogen-mediated selection heterozygote advantage, and gene flow, as reviewed in [10, 92, 93].

Selective evolutionary pressure from mate choice behaviours may be particularly strong in species with long-term pair bonds [94, 95]. Indeed, previous studies have reported some evidence of disassortative mating in regard to MHC II class in other seabirds with similar mate-choice strategies [29, 38] but see [37]. For this reason, and given Little Auk life-history traits (partners mate for multiple breeding attempts if not for life [40]), we expected to find evidence of disassortative mating but failed to do so. There could be several reasons for the absence of such a pattern. First, disassortative mating could be hindered by the overall high MHC diversity existing in Little Auks (i.e. 99 alleles across two loci), meaning that most prospective mates are already quite dissimilar at MHC II B. Second, mate choice could be operating on only one of the two characterized loci, and our inability to distinguish loci is masking an existing disassortative mating trend. Third, specific MHC genes may be only partially associated with a separate phenotypic trait valued by Little Auks, and thus mate choice may not operate on MHC disassortative principles at all. Finally, random mating occurring on two separate yet functionally expressed MHC II B loci may be sufficient to maintain the high degree of MHC

polymorphism that we find in the population, yet Little Auks may select mates on the basis of other criteria that may only loosely be correlated with MHC, such as size or experience [49].

Interestingly, we found some evidence of assortative mating. Despite the high number of MHC alleles, breeding pairs shared alleles in common more often than would be expected by chance, yet this result may be biased by the inability to distinguish between duplicated loci. However, there is a supporting rationale for non-random mating in the Little Auk that would support this result, as phenotypic mate choice decisions in the Little Auk have previously been reported to be non-random, with pairs mating assortatively in regard to some morphological traits (wing length, the size of white spot on the eye-lid [49]). Exhibiting a mate choice preference for similar MHC alleles could operate on a similar principle, or MHC alleles could covary with morphological traits (i.e. MHC genotype correlates with waddle size in pheasants [96]), which might be associated with assortative mating. While speculative, this perspective is supported in a broader ecological context: some studies suggest that assortative mating can mitigate sexual conflict over parental care [97]. If so, this could be particularly important dynamic in the case of the Little Auk, which require an extensive parental effort with a high degree of parental coordination during the incubation and provisioning periods, in order to successfully fledge their chicks [98–100].

## Conclusions

Our study found a high level of genetic diversity at the MHC II B gene in a colonial Arctic seabird, the Little Auk, and the 99 alleles we identified is one of the highest examples of MHC polymorphism described in non-passerine birds. The results agree with evolutionary theory that suggest balancing selection helps to create and maintain the high levels of polymorphism within the examined MHC region. Contrary to our expectations, we did not find evidence of disassortative mating regarding the examined MHC region, and more studies are needed to fully understand the mechanism of the mate choice in the Little Auk.

## Supporting information

**S1 Table. Sequences of primers that identify exon 2 MHC in the Little Auk not used for further genotyping as provided longer products, not appropriate for Ion Torrent sequencing.**
(DOCX)

**S2 Table. List of barcodes used in bidirectional tagging for Ion Torrent PGM sequencing.**
(DOCX)

**S3 Table. Full list of alleles with their depth and frequency in the sampled individuals.**
(DOCX)

**S4 Table. Tajima's D test statistics.**
(DOCX)

**S1 Fig. Results of randomization tests testing for MHC-based non-random mating, using average and maximum amino acid dissimilarity of PBS regions.**
(DOCX)

## Acknowledgments

We are grateful to Monika Prełowska, Mirosława Dabert and Marcin Łoś for lab assistance, and Yuki Brooknievskaya for constructive discussion, encouragement and overall support

during the preparing and submission of the manuscript. We also thank Magdalena Migalska for her help and assistance with AmpliSAT data filtering. We are also grateful to Enric Sala for his support in accessing field site and collecting samples in Franz Josef Land. We also warmly thank all fieldworkers for their help in the field. NanuTravel is acknowledged for their logistical support in East Greenland. Finally, we thank four anonymous reviewers for their suggestions that helped to improve the manuscript.

## Author Contributions

**Conceptualization:** Katarzyna Wojczulanis-Jakubas, Brian Hoover, Dariusz Jakubas, Magdalena Zagalska-Neubauer.

**Data curation:** Katarzyna Wojczulanis-Jakubas.

**Formal analysis:** Brian Hoover, Magdalena Zagalska-Neubauer.

**Funding acquisition:** Katarzyna Wojczulanis-Jakubas.

**Investigation:** Katarzyna Wojczulanis-Jakubas, Dariusz Jakubas, Jérôme Fort, David Grémillet, Maria Gavrilo, Sylwia Zielińska, Magdalena Zagalska-Neubauer.

**Project administration:** Katarzyna Wojczulanis-Jakubas.

**Visualization:** Brian Hoover, Dariusz Jakubas.

**Writing – original draft:** Katarzyna Wojczulanis-Jakubas.

**Writing – review & editing:** Brian Hoover, Dariusz Jakubas, Jérôme Fort, David Grémillet, Maria Gavrilo, Sylwia Zielińska, Magdalena Zagalska-Neubauer.

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
