## [Decision Letter · Decision Letter 0]

8 Oct 2023

PONE-D-23-28975Diversity of major histocompatibility complex of II B geneand mate choice in a monogamous and long-lived seabird, the little auk (Alle alle)PLOS ONE

Dear Dr. Wojczulanis-Jakubas,

Thank you for submitting your manuscript to PLOS ONE. After careful consideration, we feel that it has merit but does not fully meet PLOS ONE’s publication criteria as it currently stands. Therefore, we invite you to submit a revised version of the manuscript that addresses the points raised during the review process.

We look forward to receiving your revised manuscript.

Kind regards,

Tzen-Yuh Chiang

Academic Editor

PLOS ONE

Journal Requirements:

2. We noticed you have some minor occurrence of overlapping text with the following previous publication(s) among others, which needs to be addressed:

Minias, Piotr & Pikus, Ewa & Anderwald, Dariusz. (2019). Allelic diversity and selection at the MHC class I and class II in a bottlenecked bird of prey, the White-tailed Eagle. BMC Evolutionary Biology. 19. 10.1186/s12862-018-1338-3. 

Juola Frans A. and Dearborn Donald C. 2012Sequence-based evidence for major histocompatibility complex-disassortative mating in a colonial seabirdProc. R. Soc. B.279153–162 http://doi.org/10.1098/rspb.2011.0562

In your revision ensure you cite all your sources (including your own works), and quote or rephrase any duplicated text outside the methods section. Further consideration is dependent on these concerns being addressed.

3. Please include a complete copy of PLOS’ questionnaire on inclusivity in global research in your revised manuscript. Our policy for research in this area aims to improve transparency in the reporting of research performed outside of researchers’ own country or community. The policy applies to researchers who have travelled to a different country to conduct research, research with Indigenous populations or their lands, and research on cultural artefacts. The questionnaire can also be requested at the journal’s discretion for any other submissions, even if these conditions are not met.  Please find more information on the policy and a link to download a blank copy of the questionnaire here: https://journals.plos.org/plosone/s/best-practices-in-research-reporting. Please upload a completed version of your questionnaire as Supporting Information when you resubmit your manuscript.

6. We note that Figure 1 in your submission contain [map/satellite] images which may be copyrighted. All PLOS content is published under the Creative Commons Attribution License (CC BY 4.0), which means that the manuscript, images, and Supporting Information files will be freely available online, and any third party is permitted to access, download, copy, distribute, and use these materials in any way, even commercially, with proper attribution. For these reasons, we cannot publish previously copyrighted maps or satellite images created using proprietary data, such as Google software (Google Maps, Street View, and Earth). For more information, see our copyright guidelines: http://journals.plos.org/plosone/s/licenses-and-copyright.

Reviewers' comments:

Reviewer's Responses to Questions

**Comments to the Author**

1. Is the manuscript technically sound, and do the data support the conclusions?

Reviewer #1: Partly

Reviewer #2: Yes

2. Has the statistical analysis been performed appropriately and rigorously? 

Reviewer #1: Yes

Reviewer #2: Yes

3. Have the authors made all data underlying the findings in their manuscript fully available?

Reviewer #1: Yes

Reviewer #2: Yes

4. Is the manuscript presented in an intelligible fashion and written in standard English?

Reviewer #1: Yes

Reviewer #2: Yes

5. Review Comments to the Author

Reviewer #1: Wojczulanis-Jakubas and colleagues present a study on MHC variation and its impact on behavior in little auks. The study is novel in its choice of system, as the species represents a rarity in avian MHC studies that are typically restricted to easier-to-study taxa and model organisms. This species is also of interest to MHC biology for being long-lived, monogamous, and, for a portion of the year, subject to the pathogen stress from extremely high population density. The authors found high diversity at the portion of the MHC they genotyped, consistent with this high pathogen-mediated selective pressure.

The portion of the paper dedicated to MHC-based mating effects is the area with the least confidence, but that is inescapable. Although the authors of the study found no evidence for disassortative mating, this may be explained (as noted by the authors) by the auks' large effective population size and extremely high MHC polymorphism, findings that will be of interest beyond the study system. The authors did, however, find evidence of significant MHC assortative mating, but concerns about distinguishing duplicated loci reduced confidence in this result. This is a weakness of the paper, but not one that diminishes all interest in the results.

The introduction does a solid job of introducing a reader to the evolution of MHC, and why little auks in particular are of interest. Perhaps this addition would make the study overly topical, but I think mentioning highly pathogenic avian influenza (HPAI), which has been found in little auks and linked to migrating seabirds, would help non-specialists appreciate the importance of studying immune system evolution in this species.

I have concerns about the methods used to distinguish technical artifacts from real sequence variation. If these were addressed (in many cases by just providing more detail), it would increase confidence in results.

- At line 185-186, what kind of sequencing is being performed? Sanger sequencing? More detail is needed on the method.

- Can you provide more information on how you distinguished sequencing artifacts and real variation? Was there a similarity cutoff or did you have a cutoff based on coverage? Line 197: "We identified and removed two classes of artefacts created during PCR (point mutation and chimeras) and during next-generation sequencing (substitution and inserts)"

- The duplication of exon 2 in some individuals (but not all) was a point of surprise for me. Two papers (refs. 67 and 68) are given as evidence of duplication of this locus in other alcids. However, are there other auk populations with polymorphism in copy number within a population? I worry it could be a technical artifact, but I would be happy to stand corrected.

- How well did the MHC alleles sequenced via RNA match those sequenced from DNA? That is, did it look like all alleles were expressed? Did you have DNA and RNA collected from the same individual to be able to do this check?

Minor suggestions:

- I would clarify at the start of the paragraph beginning on Line 108 to make it obvious that these samples are separate from the above and represent only one population. Perhaps the following would be suitable: "To evaluate the expression status of MHC class II sequences and to identify potential pseudogenes obtained with tested primers, we [additionally] amplified and sequenced cDNA [from the spleens of 10 individuals from one population.]"

- The "Molecular sexing" paragraph occurs before any mention of DNA extraction. Please consider reordering.

- At line 141, I'm not sure what a "bench" of PCR primers is. Perhaps a "panel" would be better?

- I was surprised to see vectorette PCR used, since I thought it was not a very common technique, but I think the rationale was to design more specific primers for the species of interest? Please clarify the rationale. The paper cited as a model (ref. 50) is a Drosophila phylogeny paper from 2003. Could you explain the rationale for the selection of the method and perhaps note and cite other MHC papers that have used the same technique?

- You note ultimately the following: "We used the pair of primers: AukLz1 and AukRz3 for further MHC II B second exon genotyping. We also designed a separate set of primers (AukR0709, AukL0709 and AukL0809, AukR0809) to obtain longer sequences of the second and partial third exon of the MHC class II gene. However, the generated fragments were too long to use freely with Ion Torrent technology." So does this mean that you ended up only using the previously-published primers and that the specific primers you designed via vectorette PCR were not ultimately used? If so, please consider cutting the mention of vectorette PCR since it makes the methods section hard to understand. If you feel those primers are still worth publishing, perhaps the relevant methods can be put into a supplement and only briefly mentioned in the main text.

- At Line 196, Bioconductor is spelled incorrectly.

Reviewer #2: This paper describes a study of MHC class IIb diversity in the little auk. The authors make a good case for evaluating immunogenetic diversity in typically under-studied taxa, such as seabirds. It is important to study MHC evolution across a wide range of species so that we can get a more complete picture of the way these genes evolve across vertebrates. The methods and analyses are sound. The conclusions reached were reasonable, and the authors were careful to note the limitations of the study. I think this paper is an appropriate fit for the journal. I only have minor comments, which I will list here. Provided that the authors address all reviewer feedback, I recommend this for publication.

60: Wouldn't this scenario select for assortative *or* disassortative mate choice depending on whether the female had the desirable genotype herself?

103: should "mist-nest" be "mist-net"?

141: "was" designed

183: delete "with"

188: "amplicon"

190: "barcode"

191: delete "of"

241: delete the comma

242: What was also the case? This is vague wording. Please clarify.

246: "remaining 26 alleles"

273-275: I don't know if you need to summarize these numbers again, since you already did so in the results.

285: "such as"

314: delete "as a matter of fact"

316: What about the other species you mentioned? Do they cover the breeding grounds extensively?

321: "that are certainly worth exploring further"

Fig. 4: I'm not sure this figure needs to be in the main body of the paper because it doesn't show a significant result. Might want to add it to the supplementals.

6. PLOS authors have the option to publish the peer review history of their article (what does this mean?). If published, this will include your full peer review and any attached files.

Reviewer #1: No

Reviewer #2: No

---

## [Author Response · Author response to Decision Letter 0]

10 Dec 2023

REPLY: We prepared the manuscript and all the associated files following the indicated templates and guidance on the PLOS ONE website.

2. We noticed you have some minor occurrence of overlapping text with the following previous publication(s) among others, which needs to be addressed:

Minias, Piotr & Pikus, Ewa & Anderwald, Dariusz. (2019). Allelic diversity and selection at the MHC class I and class II in a bottlenecked bird of prey, the White-tailed Eagle. BMC Evolutionary Biology. 19. 10.1186/s12862-018-1338-3. 

Juola Frans A. and Dearborn Donald C. 2012. Sequence-based evidence for major histocompatibility complex-disassortative mating in a colonial seabird. Proc. R. Soc. B.279153–162 http://doi.org/10.1098/rspb.2011.0562

In your revision ensure you cite all your sources (including your own works), and quote or rephrase any duplicated text outside the methods section. Further consideration is dependent on these concerns being addressed.

REPLY: None of the part of the manuscript was copied from any other resources. Nevertheless we checked all the text and did not find any evidence of plagiarism. We believe that some text overlapping may be related to specific terms/definitions used often in a similar way by various authors. To minimize the issue, we modified the sentences throughout the whole manuscript. We also cited Juola and Donald (2012) in the revised version, to support some of our statements, as this is very relevant reference, while the manuscript was apparently missing it.

3. Please include a complete copy of PLOS’ questionnaire on inclusivity in global research in your revised manuscript. Our policy for research in this area aims to improve transparency in the reporting of research performed outside of researchers’ own country or community. The policy applies to researchers who have travelled to a different country to conduct research, research with Indigenous populations or their lands, and research on cultural artefacts. The questionnaire can also be requested at the journal’s discretion for any other submissions, even if these conditions are not met. Please find more information on the policy and a link to download a blank copy of the questionnaire here: https://journals.plos.org/plosone/s/best-practices-in-research-reporting. Please upload a completed version of your questionnaire as Supporting Information when you resubmit your manuscript.

REPLY: We filled the questionnaire on inclusivity in global research and submitted it along with the revised manuscript as Supporting Information, as requested.

REPLY: We clarified the inconsistencies.

REPLY: We have deposited all the data sequences in GenBank, the accession numbers: OR909382-OR909480; the sequences have been verified and will be published on Dec 13.

6. We note that Figure 1 in your submission contain [map/satellite] images which may be copyrighted. All PLOS content is published under the Creative Commons Attribution License (CC BY 4.0), which means that the manuscript, images, and Supporting Information files will be freely available online, and any third party is permitted to access, download, copy, distribute, and use these materials in any way, even commercially, with proper attribution. For these reasons, we cannot publish previously copyrighted maps or satellite images created using proprietary data, such as Google software (Google Maps, Street View, and Earth). For more information, see our copyright guidelines: http://journals.plos.org/plosone/s/licenses-and-copyright.

REPLY: We checked type of licences for maps generated in R package ggOceanMaps, used by us to create the map in Fig. 1. Two parts, i.e. lands and ice polygons are from Natural Earth, which is public domain: http://www.naturalearthdata.com. We kept it in the maps and provided this information in the figure caption. As bathymetry layer is not distributed under the CC BY 4.0 license (and actually not needed for the present study) we removed it from the map. If you need to confirm above the information about copyright, please see the ggOceanMaps package manual, section "Citations and data sources" https://mikkovihtakari.github.io/ggOceanMaps/

REPLY: We modified the resubmitted files following the instructions.

Reviewers' comments:

Reviewer #1: Wojczulanis-Jakubas and colleagues present a study on MHC variation and its impact on behavior in little auks. The study is novel in its choice of system, as the species represents a rarity in avian MHC studies that are typically restricted to easier-to-study taxa and model organisms. This species is also of interest to MHC biology for being long-lived, monogamous, and, for a portion of the year, subject to the pathogen stress from extremely high population density. The authors found high diversity at the portion of the MHC they genotyped, consistent with this high pathogen-mediated selective pressure.

The portion of the paper dedicated to MHC-based mating effects is the area with the least confidence, but that is inescapable. Although the authors of the study found no evidence for disassortative mating, this may be explained (as noted by the authors) by the auks' large effective population size and extremely high MHC polymorphism, findings that will be of interest beyond the study system. The authors did, however, find evidence of significant MHC assortative mating, but concerns about distinguishing duplicated loci reduced confidence in this result. This is a weakness of the paper, but not one that diminishes all interest in the results.

REPLY: Thank you for this comment. We totally agree with your opinion that differentiating between loci would be of great value. However, distinguishing between duplicated loci in MHC, in general, is very challenging due to the high polymorphism in this region. One of the simplest methods that needs a lot of effort and is costly is to design PCR primers that target specific MHC alleles. This is challenging mainly due to the high sequence similarity and rarely allows assign MHC alleles to specific MHC loci (Mellinger et al. 2023 PeerJ, Vekemans et al. 2021 MolEcol, Babik 2010 MolEcolRes). Therefore, the common practice in non-model animals for which MHC genes are not well recognized is to apply a different approach. Obtained fragments of DNA/RNA may correspond to specific alleles and regions within the MHC genes and therefore they are referred to the different genetic variants within the MHC region as alleles. Such variants are used as an allele proxy (and in literature such fragments are commonly called ‘alleles’). Obviously, this is not precise but gives an opportunity for a better understanding of fragment polymorphism and provides a first step to more precise analysis. Therefore, we also decided to follow this procedure.

The introduction does a solid job of introducing a reader to the evolution of MHC, and why little auks in particular are of interest. Perhaps this addition would make the study overly topical, but I think mentioning highly pathogenic avian influenza (HPAI), which has been found in little auks and linked to migrating seabirds, would help non-specialists appreciate the importance of studying immune system evolution in this species.

REPLY: Thank you for this comment. We added information about the avian influenza in the lines: 76-78.

# Relevant fragments from the manuscript:

Lines: 76-78: “Recent outbreaks of avian influenza, particularly strongly affected seabirds [35,36], highlighting the need for deeper understanding of the functioning immune system in this group”

I have concerns about the methods used to distinguish technical artifacts from real sequence variation. If these were addressed (in many cases by just providing more detail), it would increase confidence in results.

REPLY: Please, find below a more detailed piece of information on artefact filtering and workflow adapted to define putative MHC II alleles.

- At line 185-186, what kind of sequencing is being performed? Sanger sequencing? More detail is needed on the method.

REPLY: Thank you for this comment. Indeed we overlooked to specify the description of sequencing. Below, and also in the manuscript (lines 181-185, 196-197), we improved the explanation of the method we have applied. We performed PCR with our designed set of primers for MHC II and visualized PCR products on the agarose gel. Then we selected a clear, single band (PCR product) with the targeted size, cut it off from the gel, cleaned it up, and Sanger sequenced. We compared obtained sequences to the NCBI sequences resources with nblast (Nucleotide BLAST, NCBI). Thanks to that we could check and select our fragments of interest, i.e. those that aligned with bird exon 2 MHC II fragments present in the database. Sequences we obtained had some mismatches (polymorphisms) but well corresponded to little auk closely related species.

# Relevant fragments from the manuscript:

Lines 181-185: “Selected fragments (ca. 300 bp) were cut off from the gel and purified with MiniElute Gel Purification Kit (Qiagen). The fragments were Sanger sequenced with specific vectorette primers. The obtained sequences were then compared to the NCBI nucleotide resources with nblast (Nucleotide BLAST, NCBI). Based on the results, we selected fragments that aligned to exon 2 MHC II and well corresponded to the Little Auk closely related species”

Lines 196-197: “Random bands were Sanger sequenced and obtained sequences were aligned in order to confirm targeted fragment (BLAST)”

- Can you provide more information on how you distinguished sequencing artifacts and real variation? Was there a similarity cutoff or did you have a cutoff based on coverage? Line 197: "We identified and removed two classes of artefacts created during PCR (point mutation and chimeras) and during next-generation sequencing (substitution and inserts)"

REPLY: Thank you for this very important suggestion, as we probably haven’t been enough precise about this. We clarified it now in the lines: 212-227.

# Relevant fragments from the manuscript:

Lines 212-227: “Data filtering. The processing of raw data was conducted with AmpliSAT, an online wide range of tools, for amplicon analysis, also appropriate for technology that generates single-ended reads such as Ion Torrent ([58]; http://evobiolab.biol.amu.edu.pl/amplisat/index.php). From the pipeline AmpliSAT manual, we applied: AmpliCLEAN, AmpliCHECK, AmpliSAS, and AmpliCOMPARE AmpliCOMBINE tools. Therefore, we used standardised tools for identification and filtering reads, variants and amplicons according to their similarity, coverage and frequency. We removed reads with a lower average Phred quality score (lower than 30) and reads with anomalous length (too short/too long, AmpliCLEAN). To filter out the putative alleles from artefacts we used default Ion Torrent parameters: 0.5% substitution error rate, 1% indel error rate and minimum per amplicon frequency of 1%. We also removed sequencing errors [59] and chimeras (the most abundant ones) [58,60] setting 5% frequency threshold of substitution (AmpliCHECK). To compare replicated genotyping results first, we combined multiple genotyping results (AmpliCOMBINE) and then checked for variants present in one or both compared files (AmpliCOMPARE). For retrieved alleles variant depth and other statistics were summed together from both experiments, and sequencing errors based on their low frequency and/or absence in one set of individuals were removed [53,60].”

Below we also provide a bit more elaborated response.

We used the AmpliSAT, an online wide range of tools, for amplicon analysis, also appropriate for technology that generates single-ended reads such as Ion Torrent. From the pipeline manual, we applied: AmpliCLEAN, AmpliCHECK, AmpliSAS, and AmpliCOMPARE AmpliCOMBINE tools. Detailed information on the AmpliSAT workflow is provided in the document: 

http://evobiolab.biol.amu.edu.pl/amplisat/index.php?documentation, Sebastian et al. 2016). In general, AmpliSAT provides standardised tools for identification and filtering reads, variants and amplicons according to their similarity, coverage and frequency. Thus, to 

---

## [Decision Letter · Decision Letter 1]

2 Jan 2024

PONE-D-23-28975R1Diversity of major histocompatibility complex of II B geneand mate choice in a monogamous and long-lived seabird, the little auk (Alle alle)PLOS ONE

Dear Dr. Wojczulanis-Jakubas,

Thank you for submitting your manuscript to PLOS ONE. After careful consideration, we feel that it has merit but does not fully meet PLOS ONE’s publication criteria as it currently stands. Therefore, we invite you to submit a revised version of the manuscript that addresses the points raised during the review process.

We look forward to receiving your revised manuscript.

Kind regards,

Tzen-Yuh Chiang

Academic Editor

PLOS ONE

Journal Requirements:

Reviewers' comments:

Reviewer's Responses to Questions

**Comments to the Author**

1. If the authors have adequately addressed your comments raised in a previous round of review and you feel that this manuscript is now acceptable for publication, you may indicate that here to bypass the “Comments to the Author” section, enter your conflict of interest statement in the “Confidential to Editor” section, and submit your "Accept" recommendation.

Reviewer #3: All comments have been addressed

Reviewer #4: (No Response)

2. Is the manuscript technically sound, and do the data support the conclusions?

Reviewer #3: Yes

Reviewer #4: Yes

3. Has the statistical analysis been performed appropriately and rigorously? 

Reviewer #3: Yes

Reviewer #4: Yes

4. Have the authors made all data underlying the findings in their manuscript fully available?

Reviewer #3: Yes

Reviewer #4: Yes

5. Is the manuscript presented in an intelligible fashion and written in standard English?

Reviewer #3: Yes

Reviewer #4: No

6. Review Comments to the Author

Reviewer #3: The manuscript studies the diversity of MHC loci in little auks and hypothesize that disassortative mating may be present in the populations studied. Such observation would confirm a mechanism of genetic diversity maintenance for this loci complex as has been observed in other species. The authors establish the aims and background of the study in a complete and clear manner. They provide resources for future studies on this species and related. The previous comments by other reviewers have been addressed and the paper is very close to completion and acceptance for publication.

I only have one concern with the paper and it is the distinction between assortative mating and mate choice. Assortative (or disassortative) mating refers in the literature to the correlation in pairs concerning a particular trait, in this case the MHC genotypes. Mate choice, or preference, refers to the actual behaviour, males, females or both may exhibit when pairing. Correlation does not mean causation and this mean that the observation of assortative or disassortative mating does not necessarily mean there is mate choice. I suggest the authors to clarify this. For example, stating that whatever pattern of assortative or disassortative mating observed, may serve as proxy to identify the mate choice behaviour. The authors may find clarification and more in depth explanations in papers from the Emilio Rolán-Alvarez group (10.3389/fmars.2020.614237 and references therein for example).

Reviewer #4: This manuscript presents original scientific work. It deals with a very interesting and important subject of diversity of MHC class II B gene in little auk, which is the first such investigation in the species.

Although the writing is generally quite comprehensible, some parts of the manuscript are written quite unclear and sloppy, and should be rewritten. I would also suggest English editing to correct some errors and improve readability (some corrections I suggest in my comments). Additionally, there are some inconsistencies and in different sections of the manuscript (see my comments and suggestions).

Here are some comments and suggestions to improve the manuscript:

Introduction

-line 53: With this context – change to In this context, but as the next sentence has the same beginning, re-write it (use synonyms)

-line 68: As a taxon – seabirds are a taxonomically varied group (comprising of at least 4 taxonomic orders) – so this should be changed or deleted

-the authors should be consistent in writing scientific names of the species, e.g. line 77 – write Alle alle in parenthesis (and should check throughout the text)

-line 81 – delete „also“

-line 84: pathenogenic selection – should be pathogenic selection?!

-Was one of the aims to compare MHC diversity of the three sampled breeding populations? If not – why are the samples divided in three groups? If yes, I would expect more results focusing on comparing them.

Materials and methods

-the authors should write more clearly what they mean by „small blood samples“ (line 96) – I suppose they collected small volumes of blood … further, the authors should add a description of the blood collecting procedure (how did they collect blood samples from auks)

-the authors write that they collected bird samples „across most of their breeding range“ (line 97), but in the discussion section, they write that „examined individuals … do not cover the whole breeding range of the species“ (line 316) – so I find the first sentence (line 97) a bit misleading – I suggest the authors to change the first sentence

-line 108 – „To evaluate the expression status of MHC class II sequences and to identify potential pseudogenes obtained with tested primers …“ – however I did not find any results regarding to that – please add those results in the manuscript (there is mention on cDNA in discussion section, but not in the results)

-the authors further describe in detail the procedure for cDNA analysis (lines 124-137) – but it ends quite abruptly (reverse transcriptase was inactivated …) – but what succeeding analysis they did with the mixture? The authors come back to cDNA at line 178 – but it is quite difficult to follow the procedures as they are discontinuously described (maybe the authors could firstly describe primer development)

-further, it is not clear what the authors mean by (line 178) „Based on received cDNA, we performed PCR amplification in order to receive targeted sequence of MHC.“? Please, clarify.

-further, the authors mention primer aukRz1 (line 131) – here it is not clear what primer is that – at least cite Table 1 (and describe a primer – e.g. it is degenerate primer designed to anneal to ….)

-please correct to „according to manufacturer's protocol“ (lines 132, 136)

-line 140: … we used two approaches – I think it would be better to write two phases, or two steps, to procedures … as those „approaches“ followed one after another, and they were not different approaches – it is a slight difference, but I think is important to be precise to avoid misleading the reader

-line 147: … to design specific, degenerate primers …“ – I suppose they are those in Table 1 (although not all primers in Table 1 are degenerate), but it should be written more precisely … further, it is not clear why are all those primers in Table 1 important (maybe the authors could write in the text the 2 primers used for genotyping, and the rest of primers move to supplementary table) … further, use the term „degenerate primer“ instead of „degenerated primer“ throughout the text

-my suggestion would be to completely move a description of vectorette PCR to supplementary materials, so that the main text is not interrupted with such detailed description, and consequently, re-writing of that part of Mat and methods section accordingly

(it seems to me that this section is quite difficult to follow, so I suggest the authors re-write it in a clearer and more intelligible way)

-the authors should check citations in the text carefully, e.g. „Ko et al. 2003“ (lines 163, 169) – I suppose it should be citation 50

-lines 175-177: I am not sure what is the point of this sentence – is it important for this research? If it is, it should be re-written to clarify it.

-lines 185-186: what does it mean „ … in order to confirm receiving sequence of targeted MHC“?... further, the authors should write what sequencing method was used in sequencing service Genomed – and describe a bit how the sequencing was performed (e.g. primers used for sequencing)

-in relation to that – MHC genotyping (lines 188-189) is actually sequencing again – the authors should write it

-line 206: Conserved MHC class II exon 2 motifs were considered as functional alleles. – the authors should explain this

-the authors should explain better mate-choice analysis - they should explain each „criteria“ (by the way – is that appropriate expression? Later, the authors use the term „metric“, line 226), and/or cite the source for the method. Further, they only cite ii) criteria – MHC-band sharing coefficients – but the reference cites research using DNA fingerprinting (Wetton JH, Carter RE, Parkin DT, Walters D. Demographic study of a wild house sparrow population by DNA fingerprinting. Nature. 1987;327: 147–149) – so, at least the authors should explain a bit, give more details (and is the term band sharing appropriate?). Additive amino acid and maximum amino acid differences should also be explained. Further, the authors should explain how they estimated heterozygosity as they were not able to assign alleles to each of the two loci.

-lines 224-225: „We assessed amino acid substitutions for both PBR regions and total exon sequences.“ – It is not clearly written, should be explained better, re-written

-lastly, I think the authors should specify somewhere (either in Mat and methods section or in the Results) the number of individuals of each sex

Results

-line 239: … 267 bp in length fragment – earlier the length of the fragment is written as 298 bp (e.g. line 205) – it is important that this discrepancy is explained in the text (further – better to write xx bp – long fragment instead of xx bp in length fragment)

-the authors should be consistent with the naming of populations, e.g. line 247 – omit Hornsund, write only Spitsbergen – the name written on map in Fig. 1 (as for the other two populations)

-line 240: … indicating at least 2 loci (by the way – use words for numbers 1-9) and the presence of a duplicated MHC II gene fragment … - is that not the same thing? So I suggest writing „i.e.“ instead of „and“

-line 241: „On average two alleles, per individual were recorded …“ (by the way, delete the comma in this part of sentence) … but in line 244 „… 56 individuals had 2 alleles (40%)…“ – so, what does it mean „on average“ in the first sentence? Further, this is repetition … should be re-written.

-the authors should be consistent with the names of alleles (Fig 1. and Fig. 2 – allele 1, allele 2 etc., Table S2 – there is AS allele name and Allele name – both are different from those in Figure 1) – the authors should unify the names throughout the manuscript

-lines 247-251 – I would suggest that the authors write (again) the number of samples for each population, and discuss in discussion section number of alleles and number of private alleles identified per population in relation to the number of individuals analysed

-line 253: correct Table 1 to Table 2 (but if you accept my previous suggestion to move Table 1 to supplement, then this will indeed be Table 1 – of course, in that case, the authors should carefully check and correct all the tables numbers in the text)

-Table 2 – the authors should define „aa seqs“

-further, as the authors did not report basic polymorphism statistics separately for each population, I do not see the point in reporting them in the table as all those values could be written in a sentence.

-Fig 4 caption (line 706) - the authors should not repeat the explanation, they should just write what the figures depict (the explanation should be incorporated in the text of Results section)

-the authors should be consistent with writing the species name (in potential use of capital letters), e.g. in Table 2 they write Little auks, in line 172 they write Little Auk, but mostly it is written little auk – the authors should check and correct accordingly throughout the text, figures and tables

-lines 259-260: … while Tajimas D analysis revealed sites with a significant excess of high-frequency segregating sites.“ – Which sites? Where those results can be found?

-lines 261-262: First sentence is redundant.

-lines 267: Subsequent analysis suggests … - what the authors mean by „subsequent analysis“? Did they perform some subsequent analyses? If yes, should be clearly written and explained.

-line 703 (Fig 3 caption): „negatively selection“ should be corrected to „negative selection“

Discussion

-lines 281-282: Was the aim of using cDNA to confirm putative alleles? Please, refer to my previous comments on cDNA – it should be clearly noted why cDNA were used, and mentioned appropriately.

-line 285: „The high similarity between alleles, as is shown in the little auk …“ – Where is that shown in the results? It should be cited again (e.g. table yy, or by repeating the values from the results). Additionally, I suppose you meant high similarity among alleles (or you had in mind two particular alleles that were very similar)? Further, are there any other explanations for the high similarity between/among alleles?

-lines 291-292: when making comparisons or comments, the authors should explicitly specify the values that they are commenting, like here: … can reach very high number – so please specify the numbers (like they did for non-passerine birds in the next sentence). Also, specify the MHC class that you are referring to in your discussion, at least at the beginning of the sentence.

-line 295-296: the sentence is not clear, even conflicting (the little auk polymorphism is lower, but then one of the highest?) – should be re-written … further, I think the authors should put those comparisons in context of number of analysed individuals, or at least make some comment on that

-line 303: what does „then“ stand for in this sentence? What does it signify?

-line 307: „compare“ should be changed to „compared“

-line 309: „This suggests blue petrels are colonial species …“ – Does the maintenance of many MHC loci class I (but not class II) really suggest this? I think the authors should be much more attentive on the meaning of their sentences, throughout the manuscript.

-line 317: … in frequencies of the shared alleles … should be changed to …. frequencies of the most common share alleles.

-line 328: … locus -specific MHC II B genotyping was outside the focus of this study … why is it so? Could the authors comment on that in discussion?

-lines 335-336: „While we did not investigate gene expression in this study, our evidence supports the idea that the identified alleles are functional and expressed.“ Firstly, what about cDNA analysis that you wrote you performed? Secondly, what evidence supports that identified alleles are functional? You wrote „our evidence“ so the reader would expect that it is the evidence from this investigation, but then you cite some previously published papers.

-line 339: „… MHC genotypes, they are likely to be important components of little auk immune systems.“ – It is very well established that MHC genotypes are important to immunity. Is there something else the authors wanted to suggest?

-line 345: The lack of such a pattern suggests several possibilities. – This is one example of poor English writing (as far as I can tell), so the authors should consult a native speaker … maybe: There could be several reasons for the absence of such a pattern.

-lines 351-354: The sentence starts with „… random mating….“ and ends „… little auks may select mates …“ – it is quite in conflict – should be rewritten for more clarity

-lines 361-365: this is quite confusing part, it is not clearly written … „effect of the same mechanism“ – which mechanism does the authors refer to?

References

There are some errors in references, here are some of them that I noticed:

-references 33 and 34 are very similar, but cited in a different way – the authors should check them (additionally ref 34 has repetition)

-reference 38 – repetition, no journal name

-reference 39 – please check

-all references should be carefully checked once again

7. PLOS authors have the option to publish the peer review history of their article (what does this mean?). If published, this will include your full peer review and any attached files.

Reviewer #3: No

Reviewer #4: No

---

## [Author Response · Author response to Decision Letter 1]

15 Feb 2024

Dear Editors,

Following your request and taking into account comments made by Reviewers, we revised our manuscript entitled “Diversity of major histocompatibility complex of II B gene and mate choice in a monogamous and long-lived seabird, the Little Auk (Alle alle)”. Journal recommendation and our interpretation is that the comments were minor, basically to improve the clarity of the manuscript, which resulted in some changes in the text but not in findings and conclusions. We also noticed that comments of Reviewer2 have been made on the first version submitted to the journal (PONE-D-23-28975), while this version has been already corrected based on two other reviews (PONE-D-23-28975_R1). In result, some of the suggestions made by Reviewer 2 were no longer valid. Nevertheless, we thank the Reviewer for pointing out the places where we should pay particular attention to improve the text readability. We carefully considered all these suggestions, and if still relevant we corrected them accordingly. Following general recommendation and as a good practice we have also carefully proofread the whole manuscript, further clarifying the text. While doing so, we realized we should support some statements and we added three references (positions numbers in the reference list of the revised manuscript: 92, 93, 94, 97). We believe that all the changes we made greatly improved our manuscript.

We provide our detailed responses to all of the comments, after “REPLY”. We numbered all the replies, and sometimes used this numeration to refer to a previous reply. While replying and whenever relevant, we referred to the clean-revised version of the manuscript, with copy-pasted fragments of the revised manuscript. The marked manuscript with track changes is also provided, as requested. 

We are very grateful to you and the reviewers for the time and effort invested in evaluating our manuscript. We believe that our manuscript is in a good shape and we are now pleased to send you its revised version hoping that you will now find the manuscript suitable for PLOS ONE.

Best regards,

On behalf of all the co-authors,

Katarzyna Wojczulanis-Jakubas

Reviewers' comments to the Author

Reviewer #3: The manuscript studies the diversity of MHC loci in little auks and hypothesize that disassortative mating may be present in the populations studied. Such observation would confirm a mechanism of genetic diversity maintenance for this loci complex as has been observed in other species. The authors establish the aims and background of the study in a complete and clear manner. They provide resources for future studies on this species and related. The previous comments by other reviewers have been addressed and the paper is very close to completion and acceptance for publication.

I only have one concern with the paper and it is the distinction between assortative mating and mate choice. Assortative (or disassortative) mating refers in the literature to the correlation in pairs concerning a particular trait, in this case the MHC genotypes. Mate choice, or preference, refers to the actual behaviour, males, females or both may exhibit when pairing. Correlation does not mean causation and this mean that the observation of assortative or disassortative mating does not necessarily mean there is mate choice. I suggest the authors to clarify this. For example, stating that whatever pattern of assortative or disassortative mating observed, may serve as proxy to identify the mate choice behaviour. The authors may find clarification and more in depth explanations in papers from the Emilio Rolán-Alvarez group (10.3389/fmars.2020.614237 and references therein for example).

REPLY #01: Thank you for this comment, we agree with it and believe that it occurred due to imprecise wording in our text, which we have now clarified. In the manuscript, we focused on MHC-dependent mating preferences expressed as MHC-assortative/MHC-disassortative mating which we believe might serve as a proxy explanation for mate choice. We are aware that observed MHC-dependent mating preferences do not reflect mate choice per se but provide potential background for mate choice patterns. Our intention was rather to identify MHC's role in mating by exploring partners similarity/dissimilarity in a fragment of MHC. Little Auks seem to be a good model for this purpose because of their high-density colonial breeding and their exposure, therefore, to potentially high pathogen pressure (as clarified this in the manuscript, lines 86-92). Our premise is that these ecological factors should clearly predict MHC-dependent mating preferences, and subsequently, indicate whether MHC variability plays a role in mate choice decisions in the Little Auk. The general assumption that MHC plays a role in mate choice in this species, however, is paradoxically made even more complicated and interesting because the Arctic environment Little Auks inhabit is believed to present a low risk of parasitemia and pathogen exposure. In summary, we incorporated changes in the manuscript to better clarify our points above, and to be more precise in our wording about MHC-dependent mating preferences in response to the reviewer comments. Lines 81-83, 87-93.

Lines 81-83: “In this study, we focus on the Little Auk, or dovekie (Alle alle), to characterize their MHC diversity and to test whether MHC variation could be maintained in this species through MHC-dependent mating preferences. 

Lines 87-93: “Such environmental conditions could be expected to relax pathogenic selection on the MHC diversity in Little Auks. However, Little Auks also breed colonially in high densities that may facilitate pathogen transmission [27] (for example, via direct contact during sexual or aggressive interactions [47]), which in turn could increase selection pressure on MHC alleles, therefore increasing MHC diversity. These contrasting scenarios predict opposing patterns of MHC diversity in the Little Auk and complicate our understanding of the role of MHC in mating preferences and population biology of this species”

 “

Reviewer #4: This manuscript presents original scientific work. It deals with a very interesting and important subject of diversity of MHC class II B gene in little auk, which is the first such investigation in the species. Although the writing is generally quite comprehensible, some parts of the manuscript are written quite unclear and sloppy, and should be rewritten. I would also suggest English editing to correct some errors and improve readability (some corrections I suggest in my comments). Additionally, there are some inconsistencies and in different sections of the manuscript (see my comments and suggestions). Here are some comments and suggestions to improve the manuscript: 

REPLY #02: We noticed that the Reviewer’s comments have been made on the first version submitted to the journal (PONE-D-23-28975). This version has been already corrected based on two other reviews (PONE-D-23-28975_R1), so some of the suggestions are no longer valid. Nevertheless, we thank the Reviewer for pointing out the places where we should pay particular attention to improve the text readability. We considered all these suggestions, and if still relevant we corrected them accordingly. Following these and other comments, we have carefully proofread the whole manuscript and rewrote a some part of the text. We believe that the changes we incorporated according the Reviewers’ suggestions improved the manuscript.

Introduction

-line 53: With this context – change to In this context, but as the next sentence has the same beginning, re-write it (use synonyms)

REPLY #03: This part of the text has already been changed in the version of PONE-D-23-28975_R1. 

-line 68: As a taxon – seabirds are a taxonomically varied group (comprising of at least 4 taxonomic orders) – so this should be changed or deleted-the authors should be consistent in writing scientific names of the species, e.g. line 77 – write Alle alle in parenthesis (and should check throughout the text)

REPLY #04: This part of the text has already been changed in the version of PONE-D-23-28975_R1. Please see the present version: lines: 71-81.

Lines 71-81: “Due to habitat preferences and some specific life-history traits, seabirds represent an interesting group to examine the functioning of the immune system, including MHC (e.g.[23,29]). Seabirds spend most of the year in the marine environment where they are exposed to an array of specific pathogens and ectoparasites [30–32], and that may differentiate them from other avian groups. In addition, many seabirds breed colonially in dense aggregations that can promote elevated transmission rates of pathogens, and the strength of selection on MHC diversity (especially in class II genes) has been shown to increase with increased coloniality and migratory behavior in birds [33]. Recent outbreaks of avian influenza have had particularly strong effects on seabirds [34,35], highlighting the need for a deeper understanding of the functioning immune system in this group. Finally, seabirds form long-term pair bonds that are critical for the successful incubation and provisioning of the brood, and thus represent a promising avenue for MHC-dependent mate-choice studies [36–39].”

-line 81 – delete „also“

REPLY #05: This part of the text has already been changed in the version of PONE-D-23-28975_R1. 

-line 84: pathenogenic selection – should be pathogenic selection?!

REPLY #06: Corrected.

-Was one of the aims to compare MHC diversity of the three sampled breeding populations? If not– why are the samples divided in three groups? If yes, I would expect more results focusing on comparing them.

REPLY #07: We apologize for the oversight this misleading information. Our aim was to capture a MHC diversity at wider scale, so we sampled individuals from more than one breeding locations. Being aware that sampling in distant colonies potentially might show differences in MHC diversity/variation according to local habitat differences, we included information on basic differences between sampled populations (lines 109-114). However, as we primarily aimed to analyze MHC-dependent mating preferences in pairs and we have not detected significant differences in allele composition between colonies we decided not to focus on the results of the colonies’ MHC architecture. For that purpose, a greater sample size for each location would be ideal, as well as the inclusion of samples from other colonies. The present study is thus best understood as a starting point for understanding population differences in MHC diversity in the Little Auk. We corrected the paragraphs in the Introduction and Methods (lines 93-99, 121-124).

Lines 93-99: “In this study, we therefore genotyped MHC alleles across three main breeding locations (comprising two morphologically distingushable subspecies) to recognize MHC diversity of the species. Since selecting a mate is a critical decision with long-term fitness consequences in Little Auks (long-term pair bonds, social and genetic monogamy [48]; preference of partners with particular phenotypic traits [49]), and MHC II genotypes are known to mediate mate-choice decisions, we also tested the hypothesis that Little Auks exhibit MHC-dependent mating preferences. ”

Lines 109-114: “Spitsbergen and Greenland colonies are inhabited by the nominative subspecies Alle a. alle, while Franz-Josef Land is inhabited by the subspecies Alle a. polaris. The two subspecies are distinguished based on body size differences (the nominative subspecies is smaller) but are genetically very similar when assessed using neutral genetic markers [50]. The three colony locations sampled represent the most important breeding aggregations of the species (except for the Thule area in Northwest Greenland, [51]).”

Lines 121-124: “In summary, samples collected from all three colonies were used to characterize the general pattern of MHC diversity in the species, and the samples collected form the Spitsbergen colony were used to identify sex specific patterns of MHC diversity and potential MHC-dependent mating preferences within known breeding pairs.”

Materials and methods

-the authors should write more clearly what they mean by „small blood samples“ (line 96) – I suppose they collected small volumes of blood … further, the authors should add a description of the blood collecting procedure (how did they collect blood samples from auks) 

REPLY #08: We added to the sentence information on the amount of blood we have collected (lines 104-105). Blood was collected according to a standard procedure for birds. Therefore, in our opinion, there is no need to describe the standard procedure in detail. However, we have introduced small modifications the paragraph to be more precise (lines 105-106).

lines 105-106: “…we collected small blood samples (ca. 20 µL of whole blood collected from underwing vain and preserved in 90% ethanol)”

-the authors write that they collected bird samples „across most of their breeding range“ (line 97), but in the discussion section, they write that „examined individuals … do not cover the whole breeding range of the species“ (line 316) – so I find the first sentence (line 97) a bit misleading – I suggest the authors to change the first sentence

REPLY #09: We deleted “most of” as we agree it was misleading and we clarified the issue (lines 93-95, 113-114)

Lines 93-95: “we therefore genotyped MHC alleles across three main breeding locations (comprising two morphologically distingushable subspecies) to recognize MHC diversity of the species”

Lines 113-114: “The three colony locations sampled represent the most important breeding aggregations of the species (except for the Thule area in Northwest Greenland, [51])”

-line 108 – „To evaluate the expression status of MHC class II sequences and to identify potential pseudogenes obtained with tested primers …“ – however I did not find any results regarding to that – please add those results in the manuscript (there is mention on cDNA in discussion section, but not in the results)

REPLY #10: The main aim of including cDNA samples was to use them as ‘a control’ for pseudogenes. cDNA results were incorporated into the general results because there were only eight cDNA samples. Our intention was not specifically to analyze little auk cDNA but to use the samples to improve allele identification. We have now clarified this in the text, please see the lines: 129-131, 160-164, 312-315. Therefore, cDNA samples were sequenced and analyzed along with DNA samples. During allele filtering, we could check for alleles identified from cDNA and confront them with other alleles obtained from DNA samples. We would then validate the repeatability and reliability of sequencing results between two sequencing sets. Both DNA and cDNA samples provide us with information on the presence of common and rare MHC alleles. 

Lines 129-131: “We used the spleen samples to amplify and sequence complementary DNA (e.g. cDNA), in order to control and validate the expression status of MHC class II B sequences obtained from DNA samples (see the details below).”

Lines 160-164: “In the following steps of the analysis (see also below) MHC sequences obtained for cDNA samples (n = 8) were used to MHC sequences obtained from DNA samples. Based on that, we could assumed that analyzed MHC fragments represent putative, functional alleles and not non-functional pseudogenes. Therefore cDNA samples (n = 8) were used only as a control of the reliability of functionality of amplified MHC fragments.”

Lines 312-315: We did not perform any formal analysis but examining alleles from the eight individuals, we could confirm the expression status of identified MHC fragments (putative alleles), generated by primers, thus MHC diversity we quantified in the Little Auk is reliable. 

-the authors further describe in detail the procedure for cDNA analysis (lines 124-137) – but it ends quite abruptly (reverse transcriptase was inactivated …) – but what succeeding analysis they did with the mixture? The authors come back to cDNA at line 178 – but it is quite difficult to follow the procedures as they are discontinuously described (maybe the authors could f

---

## [Decision Letter · Decision Letter 2]

3 Mar 2024

PONE-D-23-28975R2Diversity of major histocompatibility complex of II B gene and mate choice in a monogamous and long-lived seabird, the little auk (Alle alle)PLOS ONE

Dear Dr. Wojczulanis-Jakubas,

Thank you for submitting your manuscript to PLOS ONE. After careful consideration, we feel that it has merit but does not fully meet PLOS ONE’s publication criteria as it currently stands. Therefore, we invite you to submit a revised version of the manuscript that addresses the points raised during the review process.

We look forward to receiving your revised manuscript.

Kind regards,

Tzen-Yuh Chiang

Academic Editor

PLOS ONE

Journal Requirements:

Reviewers' comments:

Reviewer's Responses to Questions

**Comments to the Author**

1. If the authors have adequately addressed your comments raised in a previous round of review and you feel that this manuscript is now acceptable for publication, you may indicate that here to bypass the “Comments to the Author” section, enter your conflict of interest statement in the “Confidential to Editor” section, and submit your "Accept" recommendation.

Reviewer #3: (No Response)

2. Is the manuscript technically sound, and do the data support the conclusions?

Reviewer #3: Yes

3. Has the statistical analysis been performed appropriately and rigorously? 

Reviewer #3: Yes

4. Have the authors made all data underlying the findings in their manuscript fully available?

Reviewer #3: Yes

5. Is the manuscript presented in an intelligible fashion and written in standard English?

Reviewer #3: No

6. Review Comments to the Author

Reviewer #3: MAJOR

First, I apologize to the authors for not being more careful in the first review, I placed my effort in what I have experience with (this is assortative mating and mate choice), and I did not pay enough attention to the materials and methods section. That is why my second review is considerably more extensive. Otherwise, I really like this study: the introduction, Results and Discussion parts are well written and are compelling. The methodology part, and specially the primer development section, albeit being confusing, show a lot of thought and careful consideration (as the authors showcase in the first paragraph of the discussion).

After reading the comments from the second reviewer, who seems to have confused the manuscript versions, I did pay more attention to the methodology section. The part that covers from the cDNA to the primer design is confusing. I hope that my comments can help the authors to simplify the text, make it easier to read, and give speed in addressing the issues. For this last part I have added text alternatives for the authors to consider.

#Introduction

No comments, the introduction is good.

#Materials and methods

The previous sections to the following look good to me, with perhaps the exception of the cDNA samples section (see below).

-Primer development and Ion-torrent sequencing

I sympathize with the authors concerning the inconsistency on the materials and methods section. In general, I consider the terms oligo, barcode, and primer to be used in very broad and confusing manners in the genetics research world. I think we would all beneficiate from being more descriptive in defining these.

I feel that the text must be more structured, and names used in a consistent manner. Preferably, following the same names as the literature that started using the Vectorette PCR approach (e.g. https://doi.org/10.1007/s00239-003-2510-x).

What I mean for more structure is to start with a paragraph which overviews all the steps required for the design of primers. Alternatively or additionally, the authors could draw a schematic figure which describes the steps. As an example of the first paragraph, you could use the following:

“The second exon of the MHC class II B genes, which is the … [52], was sequenced… . In order to do that, and since only one primer sequence is available for this genomic region (ref?), we used a vectorette PCR approach (see ref) to obtain adequate primers specific to the little auk. We use this approach given its succesful use in MHC research (refs). This procedure entails (1) to ligate a designed vectorette to the sticky ends of restriction enzyme digested DNA, (2) to perform a nested PCR where specific and vectorette primers are used and (3) to sanger-sequence resulting amplicons”

At this point I would like to ask why was it nested? Perhaps you should clarify that as well.

After that first paragraph you could first explain how you get the vectorette, the specific primer and the vectorette primer and then, and only then, describe the laboratory procedure. By the way, you do not seem to describe where are you getting the vectorette libraries from, and the ligation process of the vectorette to the digested cDNA is lacking. It could also be good to clarify how many different vectorette libraries you constructed.

At the end of the primer development section, I am afraid I do not understand the aim of the last paragraph. If you already have the primers that you need to amplify DNA and do ion-torrent sequencing, why are you making an extra amplification and using Sanger sequencing?

I am assuming that the genomic DNA used for the little auk primer discovery is the cDNA obtained for individuals described in the cDNA samples section. Is this correct? In that case, don’t say that they were only used as controls, it is a fundamental step in the primer discovery. The fact you can use them as controls as well is additionally. Be clear. You may have wanted to do so but in Lines 160 to 161 that quote does not make sense, were used to what?

I had to go through the literature to understand the process (I read the steps undertaken in https://doi.org/10.1007/s00239-003-2510-x), I would ask you, and specially for this kind of journal, not to assume that the readers are familiar with the concepts, and name and define things clearly and consistently through the text. For instance, if I am not getting it wrong, the primers you test from related species would be the specific primers (the primers that are known for the region of interest that you want to statistically analyze). You seem to be using interchangeably “specific”, “degenerate” and “internal”. If I do not get this wrong either, the universal primers would be what in the paper by Ko et al. (2003) are called vectorette primers (which is not the same as the vectorette itself from what I can understand), and in your case would be C20 and B21. Also, I am assuming that the digested genome you are using for primer discovery is the cDNA obtained from reverse transcription (see on the comments I made previously).

I believe that the authors have been very thorough to ensure good data quality and that the steps to reach it have been considered carefully and fully. Nonetheless, the text does not show that so well. If this section gets clearer, there is to my knowledge little else needed.

-Data filtering

No comments, this looks good.

-Genetic analysis

No comments, this looks good.

-Mate-choice analysis

This looks very good, well done.

-Results

No comments, looks good.

-Discussion

Line 379-382. I agree, very good point. And in general, very good discussion.

MINOR

Line 106. Vein not vain

Line 180. Afterwards not afterward

Line 184. You cannot start a new paragraph with “For this purpose”, the paragraph should be new to the previous one.

Line 184. I do not think that is the right reference, I looked it up and there was no mentioning of vectorette PCR.

I imagine that figures were sent separately and in better resolution. The ones present in the submitted revision do not look good.

Line 203-204. Again, consistency, are you sure they can be called vectorette primers? The vectorette primers you are using are C20 and B21, readers can get confused and assume that they are the same.

Line 360. Repeated “then”

Line 377. Nest instead of Next

Line 421-423. Can you clarify some more what you mean here?

Line 423-426. Good, but can you provide some reference for that minor correlation?

7. PLOS authors have the option to publish the peer review history of their article (what does this mean?). If published, this will include your full peer review and any attached files.

Reviewer #3: No

---

## [Author Response · Author response to Decision Letter 2]

2 Apr 2024

Dear Editors,

Following your request and taking into account comments made by Reviewer #3, we revised our manuscript entitled “Diversity of major histocompatibility complex of II B gene and mate choice in a monogamous and long-lived seabird, the Little Auk (Alle alle)”. Journal recommendation and our interpretation is that the comments were minor, basically to improve the clarity of the manuscript, which resulted in some changes in the text but not in findings and conclusions. We carefully considered all these suggestions, and corrected the manuscript accordingly. Following general recommendation and as a good practice we have also carefully proofread the whole manuscript, further clarifying the text. We believe that all the changes we made greatly improved our manuscript.

We provide our detailed responses to all of the comments, after “REPLY”. While replying and whenever relevant, we referred to the clean-revised version of the manuscript, with copy-pasted fragments of the revised manuscript. The marked manuscript with track changes is also provided, as requested. 

We are very grateful to you and the reviewers for the time and effort invested in evaluating our manuscript. We believe that our manuscript is in a good shape and we are now pleased to send you its revised version hoping that you will now find the manuscript suitable for PLOS ONE.

Best regards,

On behalf of all the co-authors,

Katarzyna Wojczulanis-Jakubas

Reviewers' comments to the Author

MAJOR

First, I apologize to the authors for not being more careful in the first review, I placed my effort in what I have experience with (this is assortative mating and mate choice), and I did not pay enough attention to the materials and methods section. That is why my second review is considerably more extensive. Otherwise, I really like this study: the introduction, Results and Discussion parts are well written and are compelling. The methodology part, and specially the primer development section, albeit being confusing, show a lot of thought and careful consideration (as the authors showcase in the first paragraph of the discussion).

REPLY #1: Thank you for finding the study well written and interesting, and thank you for you time and all the suggestion at the second round of revision. We did our best to further clarify the text, and we hope the changes are now satisfactory.

After reading the comments from the second reviewer, who seems to have confused the manuscript versions, I did pay more attention to the methodology section. The part that covers from the cDNA to the primer design is confusing. I hope that my comments can help the authors to simplify the text, make it easier to read, and give speed in addressing the issues. For this last part I have added text alternatives for the authors to consider.

REPLY #2 As stated in the previous revision (R2; responses to Reviewers comments) the feedback of Reviewer #2, for unknown for us reasons, was based on the very first version of the manuscript. After the first round of revision (R1) and then the second one (R2) the text has greatly improved. Based on the current reviewer comments we understand that still there are some issues to clarify. We did our best to further clarify the text and we hope the changes are now satisfactory. Please see below our detailed responses (with reference to the modified manuscript). 

#Introduction

No comments, the introduction is good.

REPLY #3: Thank you 

#Materials and methods

The previous sections to the following look good to me, with perhaps the exception of the cDNA samples section (see below).

REPLY #4: Please find below more information and notice that we have reorganized the section on cDNA (lines 201-217). 

Lines 201-217: “The complementary DNA (cDNA) samples were used to generate sequences which were regarded as a control of the reliability of functionality of MHC fragments amplified with designed specific primers. Fragments obtained from cDNA samples represent putative, functional alleles, not non-functional pseudogene. Therefore, in the following step we confronted MHC sequences for DNA and cDNA samples generated with use of final primers: AukLz1 and AukRz3 (Table 1). Finally, before performing the next-generation sequencing of all samples, to confirm target sequence (MHC II, exon 2 fragment), we also checked the PCR product (of DNA and cDNA samples) by Sanger sequencing in two directions with both final PCR primers. For validation, the sequenced fragments were aligned with use of the Basic Local Alignment Search Tool (BLAST).

To obtain cDNA we used RNA template (n = 8) and performed reverse transcription to generate cDNA. In the first step (pre-processing) we added 2µl of RNA template (in total around 50ng) with 0,5 µl of 10 mM primer AukRz1 (degenerate primer, Table 1) and 10,5 µl of nuclease-free treated water and incubated in 65°C for 5 min, according to manufacturer’s protocol. Then, we cooled the sample to room temperature. In the next step, we added 4 µl of 5x Reaction Buffer (part of RevertAid™ M-MuLV) to the preprocessed sample with 2 µl of mixed dNTP (10 mM, Thermo Scientific) and 1 µl of reverse transcriptase RevertAid™ M-MuLV, Thermo Scientific. We incubated the mixture for 1h at 42°C according to manufacturer’s protocol and the reverse transcriptase was inactivated for 10 min at 70°C.”

-Primer development and Ion-torrent sequencing

I sympathize with the authors concerning the inconsistency on the materials and methods section. In general, I consider the terms oligo, barcode, and primer to be used in very broad and confusing manners in the genetics research world. I think we would all beneficiate from being more descriptive in defining these. 

REPLY #5: We agree that terminology in molecular papers is sometimes difficult to follow, especially in the case of terms like a primer and an oligo. Term barcodes (or tags) are restricted to the individuals’ IDs, therefore they cannot be mixed with terms primer/oligo. We understand that this is a general Reviewer impression that has not applied to our manuscript (as we do not use the term oligo and we provided a barcode definition). In most cases, it is quite easy to navigate between these terms as usually the context is very helpful (however, probably most for someone who is familiar with technical aspects of working in a molecular lab). We did our best to clarify this whole section (please see the lines: 158-199).

Lines: 158-199: “To characterize MHC diversity in the Little Auk, we identified and sequenced the second exon of MHC class II B genes, which is the most polymorphic segment of the peptide-biding region (PBR) [52]. First, we designed and tested a set of primers (Table 1 and Supplementary materials: S1 Table), based on available sequences for closely related alcid species: the Razorbill (Alca torda) (Accession Number: EU326270.1, EU326269.1, EU326268.1), the Common Guillemot (Uria aalge) (Accession Number: EU326278.1, EU326276.1, EU326275.1EU326274.1), and the Atlantic Puffin (Fratercula arctica) (Accession Number: EU326267.1, EU326267.1, HQ822507.1, HQ822494.1). In the next step, the designed primers along with universal ones (C20, B21, [53]) were used in vectorette PCR, to obtain longer exon sequences covering the region, where we intended to locate the final primers for genotyping (Table 1 and Supplementary materials: S1 Table). In the vectorette PCR, we used total genomic DNA from the eight individuals digested with four restriction enzymes to generate vectorette-ligated DNA templates. 

For vectorette PCR, we adopted the modified vectorette PCR protocol from [53] as it allowed us to obtain information on poorly known DNA fragments of interest, and has been applied successfully in MHC research (e.g. [54–56]). In this technique, one primer (so called “internal”) is designed based on the available sequence, and the second primer is universal (C20 and B21 used as forward or reverse). This approach allows the ‘unknown’ part of the sequence to be extended in one direction and amplified (contrary to standard PCR, which requires two specific primers for amplification). Therefore, the technique provides an opportunity to design more precise, final primers for focal fragments. 

First, to construct vectorette libraries, we used ca. 5 μg portion of genomic DNA which was digested with 10 U of enzymes: MunI, XapI, EcoRI, Bsu15i REs (Fermentas) at 37°C for 5 min (other steps, including ligation, were performed according [53]). Then, we have performed vectorette PCRs, where 15 μl of reaction mixture contained 7.5 μl QIAGEN Multiplex PCR Master Mix, 0.15 μM of the specific (AukLz1 or AukLz2/AukRz1, Table 1 and Supplementary materials: S1 Table) and C20 vectorette [53] primers, and 0.5 μl of vectorette-ligated digested DNA. The PCR touchdown was as follows: 95°C/15 min, 5×(94°C/30 s, 64°C/30 s, 72°C/60 s), 5×(94°C/30 s, 60°C/30 s, 72° C/60 s), 20× (94°C/30 s, 58°C/30 s, 72°C/60 s), 72°C/3min. In the second nested vectorette we used PCR 0.5 μl of product of the first PCR as template. 30 μl of PCR mixture contained 2 μL of 10× PCR buffer with (NH4)2SO4, 2 mM MgCl2, 1 μM of each primer, 0.2 mM of each dNTP and 1 U of Taq polymerase (Fermentas). In this PCR we used one specific primer (Alal1L/Alal2R/Alal3R, Supplementary materials: S1 Table) and B21 vectorette primer [53], PCR scheme was as follows: 94°C/3 min, 30× (94°C/30 s, 58°C/30 s, 72°C/60 s), 72°C/3 min. The second PCR products were run on a 1.5% agarose gel and visible fragments (ca. 300 bp) were cut off from the gel and purified with MiniElute Gel Purification Kit (Qiagen). These fragments were Sanger-sequenced with specific vectorette primers. The obtained sequences were then compared to the NCBI nucleotide resources with nblast (Nucleotide BLAST, NCBI). Based on the sequencing results, we selected fragments that aligned to MHC II B exon 2 and corresponded to sequences known from closely related auk species. Ultimately, we were able to select final primers: AukLz1 and AukRz3 for MHC II B second exon genotyping (Table 1). The PCR product of additional pairs of primers (AukR0709, AukL0709 and AukL0809, AukR0809; Supplementary materials: S1 Table) was also identified as a region of interest, i.e. the second and partial third exon of the MHC class II gene. However, generated fragments were too long to use freely with Ion Torrent technology. ”

I feel that the text must be more structured, and names used in a consistent manner. Preferably, following the same names as the literature that started using the Vectorette PCR approach (e.g. https://doi.org/10.1007/s00239-003-2510-x).

REPLY #6: Indeed, we agree that a part of the Methods needed to be better structured. Please find this part improved (Lines 151-224; too much to copy-paste here). What is worth to notice that we include the cDNA part in a new section (named “Primers validation”, lines: 200-224). The present structure corresponds much better to the procedures we have applied and we thank the Reviewer for this idea.

We do not understand what exactly the Reviewer had in mind writing that we should follow “the same names as the literature that started using the Vectorette PCR approach”. We used consistently throughout the whole manuscript the same terminology as mentioned in the publication (which we also have cited). However, to be more precise in the present version we added “C10, B21” in two places in the text and further clarified the whole text (lines 158-199, pasted just above, under reply #5).

What I mean for more structure is to start with a paragraph which overviews all the steps required for the design of primers. Alternatively or additionally, the authors could draw a schematic figure which describes the steps. As an example of the first paragraph, you could use the following:

“The second exon of the MHC class II B genes, which is the … [52], was sequenced… . In order to do that, and since only one primer sequence is available for this genomic region (ref?), we used a vectorette PCR approach (see ref) to obtain adequate primers specific to the little auk. We use this approach given its succesful use in MHC research (refs). This procedure entails (1) to ligate a designed vectorette to the sticky ends of restriction enzyme digested DNA, (2) to perform a nested PCR where specific and vectorette primers are used and (3) to sanger-sequence resulting amplicons” 

REPLY #7: Thank you for the suggestion on structure with which we agree, as we mentioned just above (reply #6). We believe that the improved section helps to understand the workflow/procedures we have applied when designing and testing the primers. In our opinion present changes in the manuscript picture enough the handling of primers and their validation. Therefore, we do not agree that figures are needed as well as in our opinion there is no need to include more technical details. We would like to avoid unnecessarily prolonging the Method section and describing commonly used tools. Modifying the text we have attempted to balance the method description while parts that could cause potential misunderstanding we have expanded. For clarification and transparency of the methods, we have explained and developed the most important parts while for others we have provided additional references. As the manuscript concentrates on mate choice there is no reason to describe in detail methods that are not new and described elsewhere (and provided with an appropriate citation.

At this point I would like to ask why was it nested? Perhaps you should clarify that as well.

REPLY #8: We would like to note that nested PCR is a standard lab technique and we believe that there is no need to get too much into details on the techniques. The nested PCR approach eliminates spurious bands not representing the region of interest (false positive fragments), and that is obvious for researchers working with PCR techniques. Thus, if anybody would like to use our protocol, will be familiar with PCR techniques, and so there is no need to explain that in the manuscript. 

After that first paragraph you could first explain how you get the vectorette, the specific primer and the vectorette primer and then, and only then, describe the laboratory procedure. By the way, you do not seem to describe where are you getting the vectorette libraries from, and the ligation process of the vectorette to the digested cDNA is lacking. It could also be good to clarify how many different vectorette libraries you constructed.

REPLY #9: Thank you for the suggestion, however, we cannot agree with the Reviewer that we did not provide all the above-mentioned information. Nevertheless, to better refer to methods on primer development and their validation we rearrange the subchapters and at the moment the Reviewer can find all the suggested above information included in two sections “Primer development” (subchapter slightly changed) and “Primer validation” (new subchapter). 

Regarding the particular doubts (about getting the vectorette libraries, ligation and number of libraries) we have slightly modified the present version to make the information more straightforward and clear (lines 177-199). In the new subchapter (lines 201-224) we compiled information on vectorette PCR, cDNA and Sanger sequencing sections. However, all included there information was already mentioned in the previous version of the manuscript, we have mainly changed the organisation of the existing text for clarity.

Lines 177-199: „First, to construct vectorette libraries, we used ca. 5 μg portion of genomic DNA which was digested with 10 U of enzymes: MunI, XapI, EcoRI, Bsu15i REs (Fermentas) at 37°C for 5 min (other steps, including ligation, were performed according [53]). Then, we have performed vectorette PCRs, where 15 μl of reaction mixture contained 7.5 μl QIAGEN Multiplex PCR Master Mix, 0.15 μM of the specific (AukLz1 or AukLz2/AukRz1, Table 1 and Supplementary materials: S1 Table) and C20 vectorette [53] primers, and 0.5 μl of vectorette-ligated digested DNA. The PCR touchdown was as follows: 95°C/15 min, 5×(94°C/30 s, 64°C/30 s, 72°C/60 s), 5×(94°C/30 s, 60°C/30 s, 72° C/60 s), 20× (94°C/30 s, 58°C/30 s, 72°C/60 s), 72°C/3min. In the second nested vectorette we used PCR

---

## [Decision Letter · Decision Letter 3]

9 May 2024

Diversity of major histocompatibility complex of II B gene and mate choice in a monogamous and long-lived seabird, the little auk (Alle alle)

PONE-D-23-28975R3

Dear Dr. Wojczulanis-Jakubas,

We’re pleased to inform you that your manuscript has been judged scientifically suitable for publication and will be formally accepted for publication once it meets all outstanding technical requirements.

Kind regards,

Tzen-Yuh Chiang

Academic Editor

PLOS ONE

Additional Editor Comments (optional):

Reviewers' comments:

Reviewer's Responses to Questions

**Comments to the Author**

1. If the authors have adequately addressed your comments raised in a previous round of review and you feel that this manuscript is now acceptable for publication, you may indicate that here to bypass the “Comments to the Author” section, enter your conflict of interest statement in the “Confidential to Editor” section, and submit your "Accept" recommendation.

Reviewer #3: All comments have been addressed

2. Is the manuscript technically sound, and do the data support the conclusions?

Reviewer #3: Yes

3. Has the statistical analysis been performed appropriately and rigorously? 

Reviewer #3: Yes

4. Have the authors made all data underlying the findings in their manuscript fully available?

Reviewer #3: Yes

5. Is the manuscript presented in an intelligible fashion and written in standard English?

Reviewer #3: Yes

6. Review Comments to the Author

Reviewer #3: The article has achieved the requirements from review. The methods are now clear to follow and the results are of interest for the journal readers.

7. PLOS authors have the option to publish the peer review history of their article (what does this mean?). If published, this will include your full peer review and any attached files.

Reviewer #3: **Yes: **Daniel Estévez-Barcia

---

## [Editor Report · Acceptance letter]

17 May 2024

PONE-D-23-28975R3 

PLOS ONE

Dear Dr. Wojczulanis-Jakubas, 

I'm pleased to inform you that your manuscript has been deemed suitable for publication in PLOS ONE. Congratulations! Your manuscript is now being handed over to our production team.

Kind regards, 

on behalf of

Dr. Tzen-Yuh Chiang 

Academic Editor

PLOS ONE